
# Inverse modeling of GOSAT-retrieved ratios of total column $CH_4$ and $CO_2$ for 2009 and 2010.

S. Pandey[1,2], S. Houweling[1,2], M. Krol[1,2,3], I. Aben[2], F. Chevallier[4], E. J. Dlugokencky[5], L. V. Gatti[6], M. Gloor[7], J. B. Miller[5,8], R. Detmers[2], T. Machida[9], and T. Röckmann[1]

[1]Institute for Marine and Atmospheric Research Utrecht, Utrecht University, Utrecht, the Netherlands
[2]SRON Netherlands Institute for Space Research, Utrecht, the Netherlands
[3]Wageningen University, Wageningen, the Netherlands
[4]Laboratoire des Sciences du Climat et de l'Environnement (LSCE), CEA CNRS UVSQ, IPSL, Gif-sur-Yvette, France
[5]NOAA Earth System Research Laboratory, Boulder, Colorado, USA
[6]Instituto de Pesquisas Energéticas e Nucleares (IPEN), Centro de Química Ambiental, São Paulo, Brazil
[7]School of Geography, University of Leeds, Leeds, UK
[8]Cooperative Institute for Research in Environmental Sciences (CIRES), University of Colorado, Boulder, Colorado, USA.
[9]National Institute for Environmental Studies, Tsukuba, Japan

*Correspondence to:* S. Pandey (s.pandey@uu.nl)

**Abstract.**

This study investigates the constraint provided by greenhouse gas measurements from space on surface fluxes. Imperfect knowledge of the light path through the atmosphere, arising from scattering by clouds and aerosols, can heavily bias column measurements retrieved from space. To minimize the impact of such biases, ratios of total column retrieved $CH_4$ and $CO_2$

($X_{ratio}$) have been used. We apply the ratio inversion method described in Pandey et al. (2015) to retrievals from the Greenhouse Gas Observing SATellite (GOSAT). The ratio inversion method uses the measured $X_{ratio}$ as a weak constraint on $CO_2$ fluxes. In contrast, the more common approach of inverting proxy $CH_4$ retrievals (Frankenberg et al., 2005) prescribes atmospheric $CO_2$ fields and optimizes only $CH_4$ fluxes.

The TM5-4DVAR inverse modeling system is used to simultaneously optimize the fluxes of $CH_4$ and $CO_2$ for 2009 and

2010. The results are compared to proxy inversions using model-derived-$XCO_2$ mixing ratios ($XCO_2^{model}$) from CarbonTracker and MACC. The performance of the inverse models is evaluated using aircraft measurements from the HIPPO, CONTRAIL and AMAZONICA projects.

$X_{ratio}$ and $XCO_2^{model}$ are compared with TCCON retrievals to quantify the relative importance of errors in these components of the proxy $XCH_4$ retrieval ($XCH_4^{proxy}$). We find that the retrieval errors in $X_{ratio}$ (mean = 0.61 %) are generally larger than

the errors in $XCO_2^{model}$ (mean = 0.24 % and 0.01% for CarbonTracker and MACC, respectively). On the annual time scale, the $CH_4$ fluxes from the different satellite inversions are generally in agreement with each other, suggesting that errors in $XCO_2^{model}$ do not limit the overall accuracy of the $CH_4$ flux estimates. On the seasonal time scale, however, larger differences are found due to uncertainties in $XCO_2^{model}$, particularly over Australia and in the tropics. The ratio method stays closer to the *a priori* $CH_4$ flux in these regions, because it is capable of simultaneously adjusting the $CO_2$ fluxes. Over Tropical South

America, comparison to independent measurements shows that $CO_2$ fields derived from the ratio method are less realistic than





those used in the proxy method. However, the $CH_4$ fluxes are more realistic, because the impact of unaccounted systematic uncertainties is more evenly distributed between $CO_2$ and $CH_4$. The ratio inversion estimates an enhanced $CO_2$ release from Tropical South America during the dry season of 2010, which is in accordance with the findings of Gatti et al. (2014) and Vanderlaan et al. (2015).

The performance of the ratio method is encouraging, because despite the added non-linearity due to the assimilation of $X_{ratio}$ and the significant increase in the degree of freedom by optimizing $CO_2$ fluxes, still consistent results are obtained.

## 1    Introduction

Detailed knowledge of the global distribution of surface fluxes of potent greenhouse gases (GHGs) such as $CH_4$ and $CO_2$ is needed to investigate the uncertain feedback of the global carbon cycle to human disturbances. Atmospheric measurements of

these GHGs provide original information about surface fluxes. Inverse modeling methods, also known as top-down approaches, have been developed to make use of that information to obtain improved estimates of surface fluxes. Bottom-up estimates of those fluxes are used as prior values in the top-down method, and are further improved using atmospheric measurements. Inversions assimilating flask and/or *in-situ* measurements from surface networks have significantly improved our knowledge of the sources and sinks of GHGs (Bousquet et al., 2006; Bergamaschi et al., 2010; Hein et al., 1997; Houweling et al., 1999;

Peters et al., 2007; Chevallier et al., 2010; Gurney et al., 2008). However, many regions with a key role in the global annual budgets of $CO_2$ and $CH_4$ are not adequately covered by the surface measurement network. This is especially true for tropical regions and the Southern Hemisphere. Total column measurements of $CH_4$ and $CO_2$ ($XCH_4$ and $XCO_2$) by sensors onboard satellites, with near global coverage, have been used in some recent studies (Basu et al., 2013; Fraser et al., 2013; Houweling et al., 2014; Detmers et al., 2015; Basu et al., 2014).

The Greenhouse Gas Observing Satellite (GOSAT), launched in January 2009 by the Japanese Space Agency (JAXA), is the first satellite dedicated to monitoring GHGs from space (Kuze et al., 2009; Yokota et al., 2009; Yoshida et al., 2011). Onboard are the Thermal And Near Infrared Sensor for carbon Observations-Fourier Transform Spectrometer (TANSO-FTS) and a dedicated Cloud and Aerosol Imager (TANSO-CAI). TANSO-FTS measures the absorption spectra of Earth reflected sunlight in the shortwave-infrared (SWIR) spectral range, from which $XCO_2$ and $XCH_4$ are retrieved with global coverage.

Several inverse modeling studies have applied these measurements to derive constraints on the surface fluxes of $CH_4$ and $CO_2$ (Alexe et al., 2014; Basu et al., 2013; Bergamaschi et al., 2013; Fraser et al., 2013; Houweling et al., 2015; Monteil et al., 2013; Turner et al., 2015).

Systematic errors in satellite retrievals are an important factor limiting the scientific interpretation of the data, and various methods have been proposed to mitigate their impact on the inferred surface fluxes (Bergamaschi et al., 2007; Frankenberg

et al., 2005; Butz et al., 2010; Parker et al., 2015). An important source of systematic error is scattering of light by aerosols and thin cirrus clouds along the measured light path. Two retrieval methods have been developed in the past to account for atmospheric scattering, referred to as the "full-physics" and "proxy" approach. The full-physics approach tries to account for scattering-induced errors by explicitly modeling the scattering process, and retrieving scattering properties from the data (Butz





et al., 2010). The proxy method, first introduced by (Frankenberg et al., 2005), takes the ratio of $XCH_4$ and $XCO_2$ retrieved at nearby wavelengths ( 1562 to 1585 nm for $XCO_2$ and 1630 to 1670 nm for $XCH_4$) so that path length perturbations due to atmospheric scattering largely cancel out in the ratio (see equation 1). $X_{ratio}$ is multiplied with model-derived-$XCO_2$ ($XCO_2^{model}$) to derive $XCH_4$ ($XCH_4^{proxy}$) (see equation 2).

$$X_{ratio} = \frac{XCH_4^{ns}}{XCO_2^{ns}} \tag{1}$$

$$XCH_4^{proxy} = X_{ratio} \times XCO_2^{model} \tag{2}$$

Here, $XCH_4^{ns}$ and $XCO_2^{ns}$ are retrieved assuming a non-scattering atmosphere. $XCO_2^{model}$ is calculated using a transport model, normally employing $CO_2$ surface fluxes that have been optimized using surface measurements. The atmospheric $CO_2$ fields are sampled at the coordinates of the satellite measurements and converted to corresponding total columns using the retrieval-derived averaging kernels (Schepers et al., 2012).

Proxy $XCH_4$ retrievals from GOSAT have been used in many inverse modeling studies to investigate the global surface fluxes of $CH_4$ (Alexe et al., 2014; Monteil et al., 2013; Fraser et al., 2013; Bergamaschi et al., 2013). These studies rely on the assumption that the uncertainties and biases in $XCO_2^{model}$ are relatively unimportant. Some recent studies have investigated this assumption in further detail. Schepers et al. (2012) suggested that the errors in $XCH_4^{proxy}$ are mostly dominated by the errors in $XCO_2^{model}$. Pandey et al. (2015) did a series of Observing System Simulation Experiments (OSSEs) to quantify the impact of errors in $XCO_2^{model}$ on inversion-derived $CH_4$ fluxes. It was concluded that the error becomes significant when $CO_2$ fluxes are poorly constrained by the surface measurements. Parker et al. (2015) have estimated the uncertainty in $XCO_2^{model}$ by comparing values from different models. They found that the uncertainty in $XCO_2^{model}$ becomes the most important term in the error budget of $XCH_4^{proxy}$ retrieval during summer months, when the satellite instrument operates under favorable illumination conditions allowing accurate determination of $X_{ratio}$.

In an attempt to avoid the biases introduced by errors in $XCO_2^{model}$, (Fraser et al., 2014) developed the 'ratio' method, which simultaneously constrains $CO_2$ and $CH_4$ fluxes by assimilating $X_{ratio}$ on the sub continental scale using the ensemble Kalman filter. Pandey et al. (2015) also developed a similar ratio inversion method for jointly optimizing the surface fluxes of $CH_4$ and $CO_2$ on the model grid scale using a variational optimization method. Fraser et al. (2014) compared posterior $CH_4$ and $CO_2$ flux uncertainties derived from a ratio inversion with traditional $CH_4$ proxy and $CO_2$ full-physics inversions and reported a larger reduction in uncertainty than the two in the tropics for the fluxes of both tracers.

This study extends the work of Pandey et al. (2015), by separately inverting real GOSAT measurements of $X_{ratio}$ and $XCH_4^{proxy}$ in a consistent and comparable framework to investigate the following questions: 1) How do errors in $XCO_2^{model}$ influence the results of a $XCH_4^{proxy}$ inversion? 2) How does the $X_{ratio}$ inversion system developed by Pandey et al. (2015) perform using real data? The performance of the inversions is evaluated using independent aircraft measurements. We provide an estimate of the posterior uncertainties of the $X_{ratio}$ inverted fluxes using the Monte-Carlo method described by Chevallier et al. (2007).





This paper is organized as follows. The following section explains the methods used in this study. Subsection 2.1 describes the inverse model and the *a priori* flux assumptions. Subsection 2.2 lists the measurements that are assimilated in the inversions and used for validation. Subsection 2.3 provides an overview of the inversions performed in the study. Section 3 presents the inversion results and Section 4 discusses their implications for the use of satellite retrievals in inversion studies. Finally, we
give the overall conclusions of this work.

## 2   Method

We invert GOSAT-retrievals of $X_{ratio}$, and $XCH_4^{proxy}$, each together with flask-air $CH_4$ and $CO_2$ measurements from the NOAA Global Greenhouse Gas Reference Network (GGGRN) to provide monthly surface fluxes of $CO_2$ and $CH_4$ using the TM5-4DVAR inversion system (Meirink et al., 2008). This is done as follows:

1. GOSAT-retrieved total column measurements of $X_{ratio}$ are compared to measured ratios of $XCH_4$:$XCO_2$ from the Total Carbon Column Observing Network (TCCON) of ground based sun-tracking Fourier Transform Spectrometers (FTSs) (Wunch et al., 2011).

2. GOSAT $X_{ratio}$ measurements are bias corrected by fitting a linear function of surface albedo to the residual differences between GOSAT and TCCON. This done in $X_{ratio}$ space.

3. GOSAT $X_{ratio}$ measurements are multiplied by $XCO_2^{model}$ to generate $XCH_4^{proxy}$ measurements. Two different versions of $XCO_2^{model}$ are used [see Section 2.2] to investigate the sensitivity to model errors.

4. The $XCH_4^{proxy}$ and $X_{ratio}$ measurements are inverted along with surface observations and the resulting posterior surface fluxes are integrated over the TRANSCOM regions (see supplementary Figure 4).

5. The posterior flux uncertainty for all inversions is quantified using a Monte-Carlo approach (see Appendix B) for con-
sistent comparison.

6. The performance of the inversions is evaluated and compared using independent aircraft measurements.

The remainder of this section explains these steps in further detail.

### 2.1   Inversion setup

$CH_4$ fluxes are optimized as a single flux category, representing the sum of all processes. For $CO_2$, biospheric and oceanic fluxes
are optimized separately. The *a priori* $CH_4$ fluxes used in the study are the same as used in Houweling et al. (2014), except for the Anthropogenic emissions. We use the 4.2FT2010 versions of EDGAR (European Commission, Joint Research Centre (JRC)/Netherlands Environmental As- sessment Agency), whereas Houweling et al. (2014) uses 4.1 version (http://edgar.jrc. ec.europa.eu). The *a priori* $CO_2$ fluxes come from CarbonTracker, CT2013B Peters et al. (2007), in which biosphere fluxes are based on the Carnegie-Ames-Stanford Approach (CASA) biogeochemical model (CASA), fire fluxes are based on Global Fire



Emissions Database v3.1 (GFED) and ocean fluxes are based onJacobson et al. (2007). Fossil fuel emissions in CarbonTracker are based on the Miller module (http://www.esrl.noaa.gov/gmd/ccgg/carbontracker/CT2013B_doc.php#tth_sEc5) The *a priori* flux covariance matrix is constructed assuming relative flux uncertainties of 50%, 84% and 60% per grid box and month for the total $CH_4$, biospheric $CO_2$, and oceanic $CO_2$ categories, respectively. The fluxes are assumed to be correlated temporally

using an exponential correlation function with temporal scales of 3, 3, and 6 months, respectively, and spatially with Gaussian functions using corresponding length scales of 500, 500 and 3000 km for total $CH_4$, biospheric $CO_2$, and oceanic $CO_2$, respectively.

## 2.2 Measurements

Here we give a brief account of the measurements that were assimilated (GOSAT and NOAA) or used for validation (TCCON

and aircraft-measurements).

### 2.2.1 GOSAT

The $XCH_4^{ns}$ and $XCO_2^{ns}$ terms in equation 1 were taken from the RemoTec $XCH_4$ Proxy retrieval v2.3.5 . More information about the dataset can be found in Product User Guide on the ESA GHG CCI website (www.esa-ghg-cci.org/?q=webfm_send/180) The RemoTeC algorithm uses GOSAT TANSO-FTS NIR and SWIR spectra to retrieve simultaneously $XCH_4^{ns}$ and $XCO_2^{ns}$

assuming a non-scattering atmosphere (Schepers et al., 2012). $X_{ratio}$ values were translated into $XCH_4^{proxy}$ using $XCO_2^{model}$ derived from the following: 1. Monitoring Atmospheric Composition and Climate (MACC) Reanalysis $CO_2$ product (www.copernicus-atmosphere.eu). It uses Laboratoire de Météorologie Dynamique transport model (LMDZ) (Chevallier, 2013). The corresponding $XCH_4^{proxy}$ product will be referred as $XCH_4^{ma}$. 2. CarbonTracker-2013B (http://www.esrl.noaa.gov/gmd/ccgg/carbontracker/). These $CO_2$ fields are calculated using the TM5 model as used in this study. The corresponding $XCH_4^{proxy}$

product will be referred to as $XCH_4^{ct}$.

Both data assimilation systems optimized the $CO_2$ fluxes using surface measurements of $CO_2$. For GOSAT measurements, we only used the high-gain soundings from GOSAT under cloud free conditions from nadir mode. This was done to avoid any systematic inconsistency among the operation modes of TANSO. Figure 1 shows the spatial coverage of the GOSAT dataset used in our inversions. Systematic mismatches between NOAA-optimized and GOSAT-optimized TM5 $CH_4$ fields

were observed by Monteil et al. (2013). We apply an additional bias correction to $X_{ratio}$ and $XCH_4^{proxy}$ by comparing them to total column $CH_4$ and $CO_2$ optimized via an inversion using TM5-4DVAR and NOAA flask-air data (see Appendix A).

### 2.2.2 TCCON

TCCON is a global network of ground-based FTS instruments, for measuring the total column abundance of several gases, including $XCO_2$ and $XCH_4$, in the near nfrared region of the electromagnetic spectrum (Wunch et al., 2011). These mea-

surements are the standard for validating total column retrievals from greenhouse gas observing satellites such as GOSAT. We validate $XCH_4^{ns}$, $XCO_2^{ns}$, $X_{ratio}$, $XCO_2^{ma}$, $XCO_2^{ct}$ with corresponding values of $XCH_4$, $XCO_2$ and $XCH_4$:$XCO_2$ measured



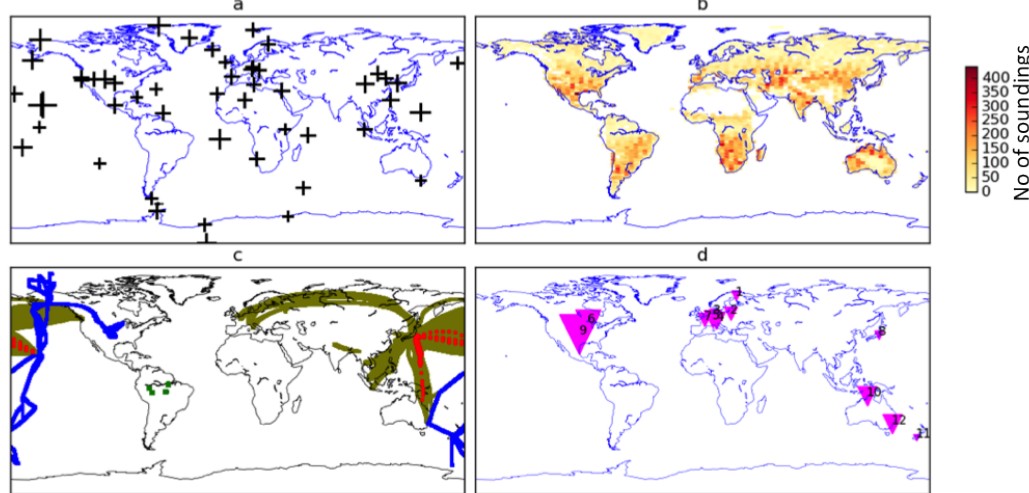

**Figure 1.** Measurements used in this study. a) The crosses indicate the locations of NOAA/ESRL surface sampling sites. The lengths of the vertical and horizontal bars are proportional to the number of $CO_2$ and $CH_4$ measurements, respectively. b) The number of GOSAT soundings binned at $1° \times 1°$ for the time period of June 2009 to December 2010; c) Flight tracks of the aircraft campaigns HIPPO 2 and 3 (blue), CONTRAIL $CO_2$ (olive), CONTRAIL $CH_4$ (red), and AMAZONICA (green); d) The locations of the TCCON measurement sites. The numbers (1- 12) refer to corresponding entries in Table 1. The size of the purple rectangles is proportional to the number of collocated high-gain GOSAT soundings

by TCCON at 12 sites using the GGG2014 release of TCCON dataset (see Figure 1 and section 3.1). An albedo-based bias correction was applied to GOSAT-retrieved $X_{ratio}$ to account for mismatch with TCCON $X_{ratio}$. (see Appendix A).

### 2.2.3 NOAA

High accuracy surface measurements of $CH_4$ and $CO_2$ were used from NOAA's GGGRN (http://www.esrl.noaa.gov/gmd/ccgg/ index.html ). The standard scales used for $CO_2$ is the WMO X2007 scale and for $CH_4$ is WMO X2004 scale. Only the sites with continuous data coverage (on a roughly weekly basis) without gaps in the time period of 1 June 2009 to 31 Dec 2010 were included. A total of 8552 $CH_4$ observations and 7843 $CO_2$ observations were used from the same 51 sites. Figure 1 shows the location of the observation sites. 1 $\sigma$ uncertainties of 0.25 ppm and 1.4 ppb were assigned to $CO_2$ and $CH_4$ measurements, respectively (Basu et al., 2013; Houweling et al., 2014). Note that our system also assigns modeling errors to each observation, depending on simulated local gradients in mixing ratio (Basu et al., 2013). Modeling error values have a mean of 27.5 ppb, 2.72 ppm ( and 1 $\sigma$ of 25.5 ppb, 4 ppm) for $CH_4$ and $CO_2$, respectively.

### 2.2.4 Aircraft Measurements

Airborne measurements from various aircraft measurement projects were used to test the inversion optimized model (see section 3.2.5). The following projects have been used:





1. HIAPER Pole-to-Pole Observations (HIPPO) from Wofsy et al. (2012a).

2. Comprehensive Observation Network for TRace gases by AIrLiner (CONTRAIL) from Machida et al. (2008).

3. IPEN aircraft measurements over Brazil (referred as AMAZONICA) from Gatti et al. (2014).

HIPPO provides *in-situ* measurements covering the vertical profiles of $CO_2$ and $CH_4$ over the Pacific spanning a wide range
in latitude (approximately pole-to-pole), from the surface up to the tropopause. We used data from the HIPPO 2 (October
26, 2009 to December 19, 2009) and HIPPO 3 (March 20, 2010 to April 20, 2010) campaigns. The continuous *in-situ* mea-
surements of $CH_4$ and $CO_2$ that were used have been bias corrected with flask air samples that were collected during each
flight and analyzed at NOAA (Wofsy et al., 2012b).This allows us to make consistent comparison with our inversions models,
as all of them assimilate NOAA flask measurements. CONTRAIL makes use of commercial airlines to measure *in-situ* $CO_2$
by continuous measurement equipment (Machida et al., 2008). For some of the CONTRAIL flights $CH_4$ measurements are
also available from flask-air samples. We use data from a lower-troposphere greenhouse-gas sampling program as part of the
AMAZONICA project, over the Amazon Basin in 2010, measuring bi-weekly vertical profiles of $CO_2$ and $CH_4$ from above
the forest canopy to 4.4 km above sea level at four locations: Tabatinga (TAB), RioBranco (RBA), Alta Floresta (ALF), and
Santarem (SAN) (Gatti et al., 2014). The coverage of all aircraft measurements that were used in this study is shown in Figure
15   1.

### 2.3   Inversion Experiments

The following inversions have been performed:

1. SURF: Inversions assimilating flask air measurements of $CH_4$ or $CO_2$ to constrain surface fluxes of $CH_4$ or $CO_2$, re-
spectively.

2. RATIO: Inversion assimilating $X_{ratio}$ and flask air measurements of $CH_4$ and $CO_2$ to constrain surface fluxes of $CH_4$ and
$CO_2$.

3. PR-MA: Inversion assimilating proxy $XCH_4^{ma}$ and flask air measurements of $CH_4$ to constrain surface fluxes of $CH_4$.

4. PR-CT: Inversion assimilating proxy $XCH_4^{ct}$ and flask air measurements of $CH_4$ to constrain surface fluxes of $CH_4$.

To assess the relative performance of each inversion, we validate atmospheric concentrations as simulated using the opti-
mized fluxes from the different inversions with aircraft measurements. We define a normalized (i.e., divided by $n$) chi-square
statistic to quantify the agreement between the optimized model and aircraft measurements.

$$\kappa = \frac{1}{n}(\boldsymbol{y} - \boldsymbol{Hx})^T \mathbf{R}^{-1}(\boldsymbol{y} - \boldsymbol{Hx}), \tag{3}$$

Where $\boldsymbol{y}$ is a vector of the aircraft measurements, $n$ is the length of $\boldsymbol{y}$. $\boldsymbol{Hx}$ is the TM5 simulation sampled at the measurement
coordinates. The covariance matrix $\mathbf{R}$ represents the expected uncertainty in the model–data mismatch. Its diagonal elements





are calculated as the sum of the model representation error of TM5 and the measurement uncertainty; all non-diagonal elements
are zero.

## 3   Results

### 3.1   GOSAST-TCCON comparison

TCCON measurements are used to investigate the errors in GOSAT-retrieved $XCH_4$. Each term on the right hand side of
equation 2 contributes to the uncertainty in $XCH_4^{proxy}$. To quantify these error contributions, we compare TCCON measurements
of $X_{ratio}$, $XCH_4$ and $XCO_2$ to corresponding co-located GOSAT-retrievals . The validation is carried out for the time period of
1st June 2009 to 31st December 2013, for which both proxy datasets ($XCH_4^{ma}$ and $XCH_4^{ct}$) are available. Table 1 shows mean
differences per TCCON station, expressed as fractional differences to facilitate the comparison of quantities with different
units. As expected, the largest differences between GOSAT and TCCON are found for $XCO_2^{ns}$ and $XCH_4^{ns}$. In general, $XCO_2^{ns}$
(mean= -1.57%) shows larger relative differences than $XCH_4^{ns}$ (mean =-0.95%). A latitudinal dependence can be observed,
with increasing biases towards stations at higher latitudes. This can be explained by increased aerosol scattering at larger sun
angles, as the light path through the atmosphere is longer. For all the stations, the mean difference is negative which is expected
for aerosol scattering-induced errors at the low surface albedos of the TCCON sites (Houweling et al., 2004). The smaller bias
values for $X_{ratio}$ than $XCO_2^{ns}$ and $XCH_4^{ns}$ confirm that scattering-induced errors cancel out in the ratio, which motivated the
proxy approach (Frankenberg et al., 2005). Overall, we observe that $X_{ratio}$ (mean bias = 0.59 %) is the dominant contributor
to the error in $XCH_4^{proxy}$. MACC ($XCO_2^{ma}$, mean bias =0.01%) and CarbonTracker ($XCO_2^{ct}$, mean bias =0.24%) contribute
relatively less errors.

### 3.2   Inversion results

#### 3.2.1   Assimilation statistics

Figure 2 summarizes the statistics of the model – measurement comparison. The prior $X_{ratio}$ mismatches typically fall in the
range -1% to +1% (with mean = 0.007 ppb/ppm and 1 $\sigma$ = 0.043 ppb/ppm). The inversions reduce the average mismatch by
about a factor of 10, and the variation of single column mismatches by about a factor of 2. The $XCH_4^{proxy}$ of PR-CT and PR-MA
have bimodal prior mismatches, because the *a priori* model overestimates the north-south gradient of $CH_4$. The bottom panels
of Figure 2 show mismatches between TM5 and surface flask measurements of $CH_4$ and $CO_2$. The $CH_4$ *a priori* measurement
mismatch has a mean of -18.30 ppb and a 1 $\sigma$ of 42.30 ppb. The RATIO, SURF, PR-CT, and PR-MA inversions are all able to
fit the NOAA data to a similar extent, reducing the *a priori* differences by more than a factor of 20. $CO_2$ flask measurements
are assimilated in SURF and RATIO. Both inversions reduce the *a priori* mismatch (mean = -2.12 ppm, 1 $\sigma$ = 3.88 ppm), with
RATIO (mean = -0.04 ppm, 1 $\sigma$ = 3.69 ppm) fitting the $CO_2$ flask data as well as SURF (mean = -0.06 ppm, 1 $\sigma$ = 3.72 ppm).





**Table 1.** TCCON validation of the components of $XCH_4^{proxy}$ (see Equation 2). The numbers represent mean percentage differences with TCCON (weighted with TCCON + GOSAT error). A negative number means that the satellite retrieval is lower than TCCON. Data from these stations was used: Sodankyla (Kivi et al., 2014), Bialystok Deutscher et al. (2014a), Bremen (Deutscher et al., 2014b), Garmisch (Sussmann and Rettinger, 2014), Karlsruhe (Hase et al., 2014), Parkfalls (Wennberg et al., 2014a), Orleans (Warneke et al., 2014), Tsukuba (Morino et al., 2014), Lamont (Wennberg et al., 2014b), Darwin (Griffith et al., 2014a), Lauder (Sherlock et al., 2014), Wollongong (Griffith et al., 2014b) are arranged from north to south. (for TCCON site locations see Figure 1).

| Station | No. of collocated measurements | $XCO_2^{ns}$ | $XCH_4^{ns}$ | $X_{ratio}$ | $XCO_2^{ct}$ | $XCO_2^{ma}$ |
|---|---|---|---|---|---|---|
| Sodankyla | 434 | -3.03 | -2.81 | 0.21 | 0.68 | 0.34 |
| Bialystok | 731 | -2.46 | -1.79 | 0.62 | 0.31 | 0.05 |
| Bremen | 426 | -1.81 | -0.99 | 0.76 | -0.05 | -0.32 |
| Garmisch | 1295 | -1.93 | -1.09 | 0.76 | 0.40 | 0.05 |
| Karlsruhe | 1244 | -1.74 | -1.00 | 0.69 | 0.15 | -0.25 |
| Parkfalls | 2174 | -1.23 | -0.43 | 0.75 | 0.22 | 0.11 |
| Orleans | 808 | -1.53 | -0.75 | 0.75 | 0.21 | -0.09 |
| Tsukuba | 135 | -1.87 | -1.24 | 0.63 | 0.57 | -0.03 |
| Lamont | 5617 | -0.73 | -0.03 | 0.68 | 0.07 | -0.00 |
| Darwin | 1065 | -0.67 | -0.19 | 0.47 | 0.02 | 0.14 |
| Lauder | 110 | -1.05 | -0.57 | 0.46 | 0.14 | 0.03 |
| Wollongong | 1515 | -0.81 | -0.45 | 0.35 | 0.10 | 0.00 |

### 3.2.2 CH$_4$ fluxes

Optimized annual CH$_4$ fluxes, integrated over the TRANSCOM regions are shown in the left panel of Figure 3. The fluxes obtained with the RATIO inversion are on average more similar to fluxes from other GOSAT inversions than to the surface inversion, with a few exceptions. Differences between satellite and surface inversion are most prominent over Tropical South

5  America, where the latter is closer to the prior, which can likely be explained by the lack of surface measurement coverage. We will return to the inversion results for Tropical South America in section 3.2.6, where validation results are shown using aircraft data.

    The most significant difference between the satellite inversion and SURF is found for Temperate Eurasia, where SURF reduces the CH$_4$ emissions from 121 Tg/y in the prior estimate to 66 Tg/y. When satellite data are added, the fluxes increase again

10  to 100 Tg/y in the region. The large flux correction in the SURF inversion is compensated by increases in other TRANSCOM regions of 5-10 Tg/y (see for example Temperate North and South America). In those regions satellite inversions remain closer to the prior than the SURF inversion, which may well be driven by the much smaller flux corrections for Temperate Eurasia. The exception is Europe, where the satellite inversions show larger reductions of up to 15 Tg/y. The large adjustments over Temperate Eurasia are analyzed further in section 3.2.7. PR-CT and PR-MA result in relatively similar posterior annual fluxes



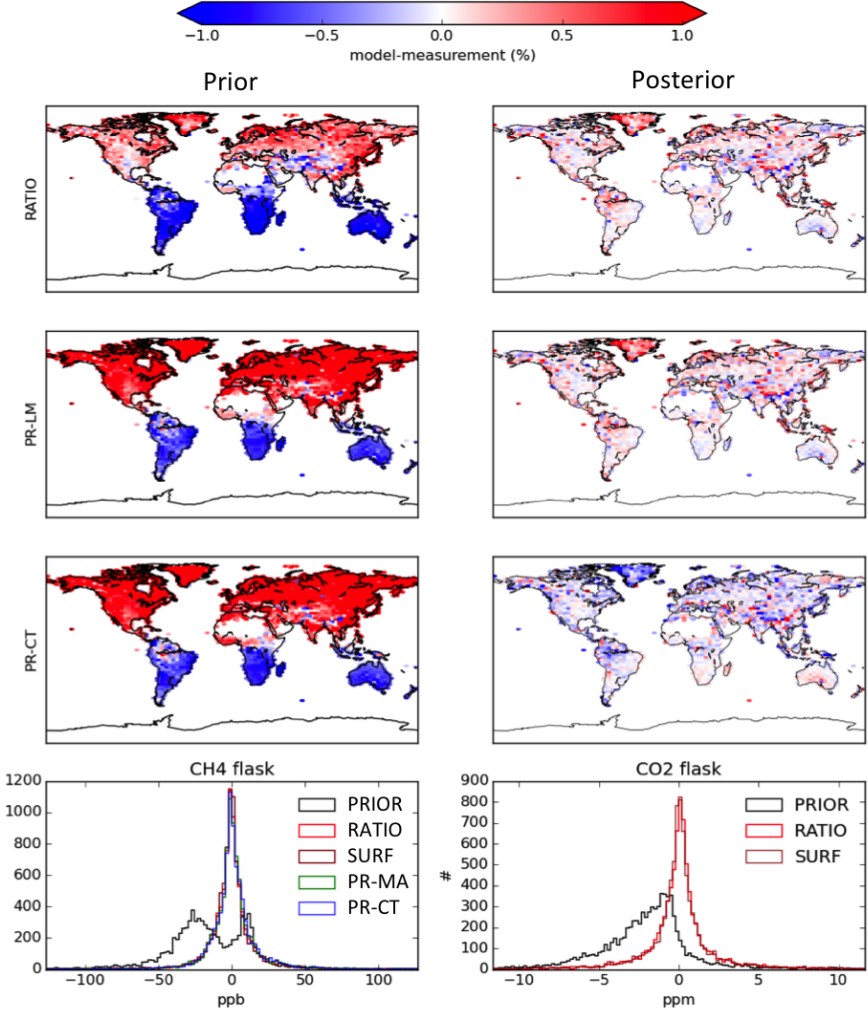

**Figure 2.** Fit residuals, comparing the performance of different inversions. The top three rows show the difference between TM5-4DVAR and GOSAT measurements ($X_{ratio}$ for RATIO, $XCH_4^{proxy}$ for PR-CT and PR-MA), using *a priori* (left) and *a posteriori* (right) fluxes. The bottom row shows histograms of measurement–model mismatches between TM5-4DVAR and NOAA surface measurements in 400 bins between $\pm 10\,\sigma$ range of the *a priori* mismatch.

for all regions. RATIO is in good agreement with the proxy inversions except for Tropical South America and Southern Africa. The right panel of Figure 3 shows annual fluxes integrated over large regions on the globe. We find a consistent adjustment in the north-south gradient of $CH_4$ compared to the prior in all inversions, corresponding to an emission shift from the Northern to the Southern Hemisphere of approximately 50 Tg/y. This might be due to an overestimation of the *a priori* emissions from





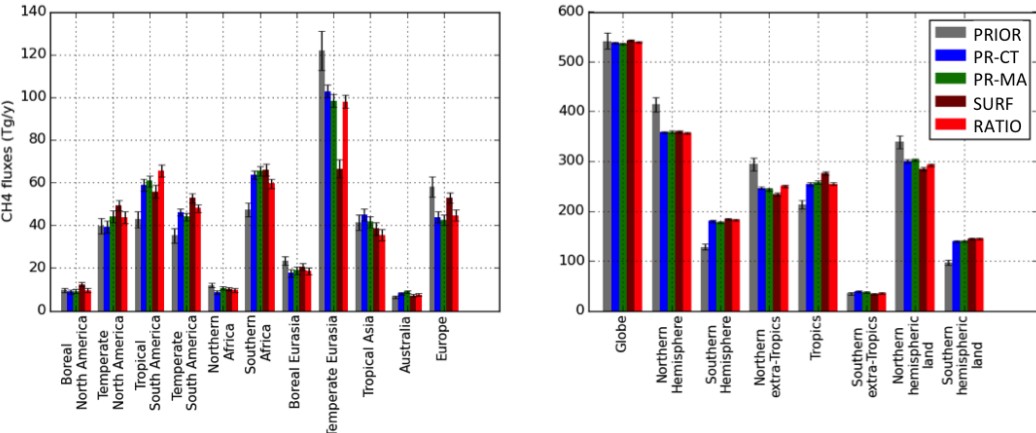

**Figure 3.** Annual fluxes of $CH_4$ integrated over different regions. The black line on each bar represents the +- 1 $\sigma$ uncertainty.

northern wetlands, as discussed in (Spahni et al., 2011). A bias in inter-hemispheric transport in TM5 is not a likely cause, since the use of ECMWF archived convective fluxes in TM5 has been shown to lead to a realistic simulation of the north-south gradient of SF6 (Vanderlaan et al., 2015). Houweling et al. (2014) found similar $CH_4$ emission shifts between the hemispheres, after bringing the inter-hemispheric transport in agreement with SF6 using a parameterization of horizontal diffusion.

Next we shift focus to seasonal differences between the inversion-derived methane fluxes (see Figure 4). Also on the seasonal scale, RATIO resembles the two PROXY inversions more than SURF. In Boreal North America, the satellite inversions that assimilate GOSAT soundings are in better agreement with the prior. We observe an increase in summertime $CH_4$ fluxes in SURF estimates for Boreal and Temperate North America. The differences in annual mean fluxes discussed earlier for Tropical South America and Temperate Eurasia do not show an important seasonal dependence. Large differences in seasonality are

obtained for Australia and the African regions, which also show important differences between the two proxy inversions (see Section 3.2.4). In Southern Africa, all inversions show increased $CH_4$ fluxes compared to the prior estimate; however, small differences can be seen between the two proxy inversions, especially in 2010. SURF remains in good agreement with PRIOR, which is expected as no surface observations are available to constrain the fluxes in this region.

### 3.2.3    CO$_2$ fluxes

Annual $CO_2$ fluxes from the SURF and RATIO inversions, integrated over TRANSCOM regions, are shown in Figure 5. Overall, we find good consistency between the results from RATIO and SURF except for Temperate Eurasia, where RATIO results in a higher $CO_2$ uptake of 0.5 PgC/y. Corresponding reductions in $CH_4$ fluxes are found for this region in the RATIO inversion. This can be understood by realizing that the satellite information, that is used, consists of the ratio of $CH_4$ and $CO_2$ columns. A RATIO inversion can simultaneously reduce the $CO_2$ and $CH_4$ fluxes over a region without changing the $X_{ratio}$ in

the atmosphere. SURF points towards a natural sink of 0.5 PgC/y in Boreal North America. RATIO and the *a priori* are carbon neutral in this region. This agreement is also seen on the $CH_4$ side of the RATIO inversion. Only small differences between




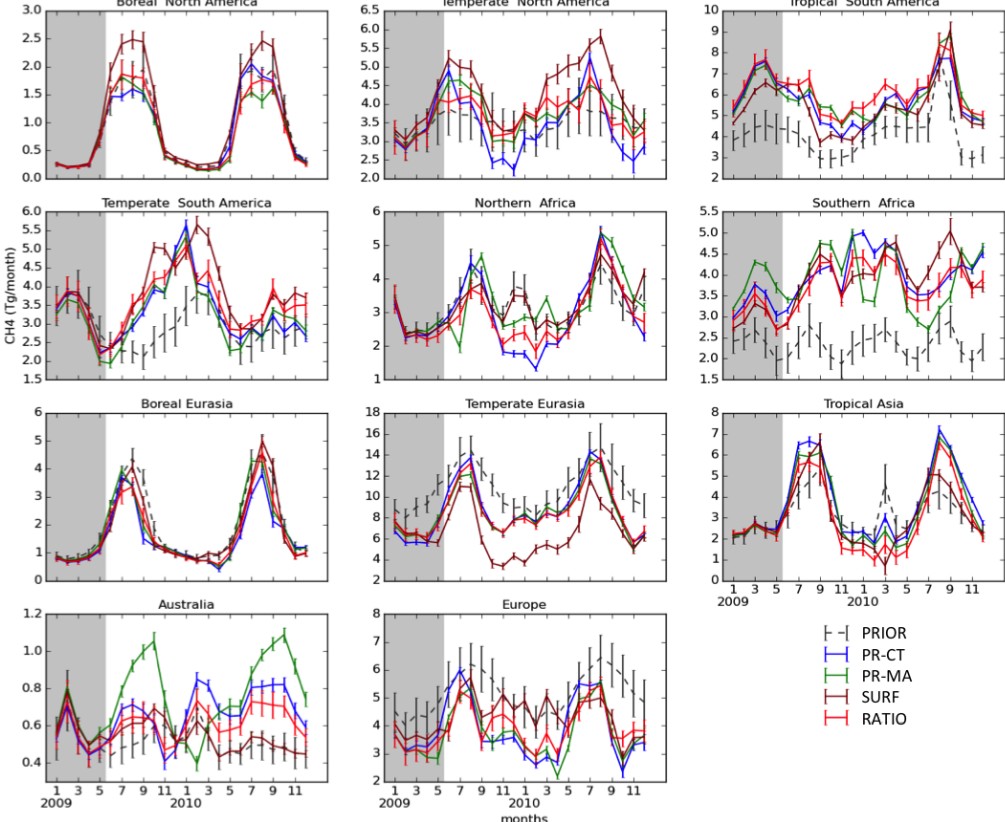

**Figure 4.** Monthly fluxes of $CH_4$ integrated over TRANSCOM regions. The vertical lines represent a 1 $\sigma$ uncertainty of the monthly fluxes. The gray region in each plot represents the period in which no measurements are assimilated.

the posterior and prior fluxes of SURF and RATIO are found over the oceans except for the Temperate North Pacific, which is neutral in both inversions compared to a sink of -0.5 PgC/y in the prior fluxes, and in Tropical India which is turned into a net sink. Interestingly, RATIO leads to posterior fluxes for Europe that are close to carbon neutral for the analysis period. This is in contrast with the findings of several inversions using GOSAT full physics $XCO_2$ retrievals, suggesting a largely underestimated
5   European carbon sink of the order of 1 PgC/y (Basu et al., 2014; Chevallier et al., 2014; Reuter et al., 2014; Houweling et al., 2015).

The RATIO and SURF inversions increase the global $CO_2$ sink of the terrestrial biosphere compared with the *a priori* fluxes. This is primarily caused by the bottom-up CASA model, which has been reported to underestimate the carbon uptake of the Northern biosphere sink in summer season (Yang et al., 2007).Basu et al. (2013) also find a global natural sink of 3 to 4 PgC/y
10  for GOSAT and NOAA inversions. This natural sink is needed to fit the atmospheric growth rate of $CO_2$ in the presence of about 9 PgC/yr anthropogenic emissions. The Southern Hemisphere land is turned into a source of 1 PgC/y in both inversions.





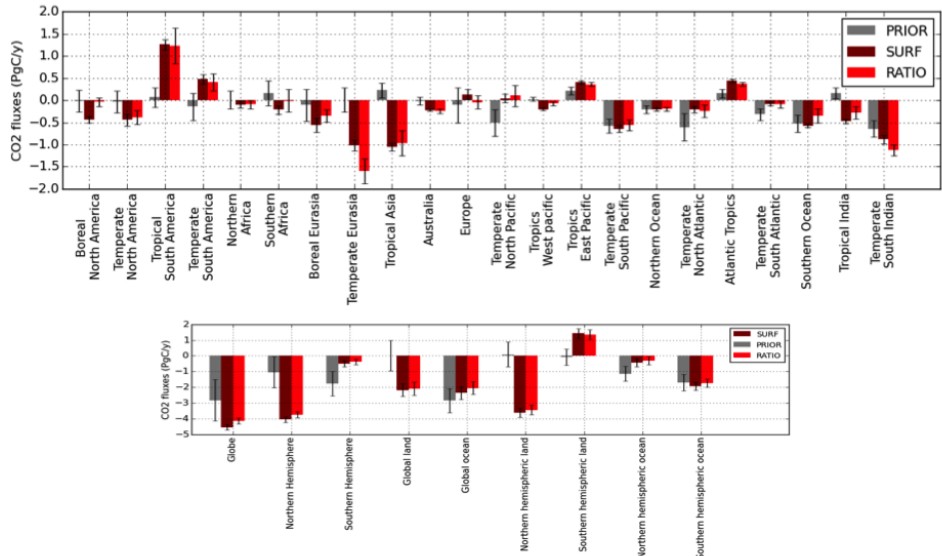

**Figure 5.** Annual fluxes of $CO_2$ (excluding fossil fuel emissions) integrated over different regions.

### 3.2.4 Errors in $CO_2^{model}$

In this Section, we analyze the differences between the two proxy retrievals ($XCH_4^{ct}$ and $XCH_4^{ma}$) and how they propagate into posterior $CH_4$ fluxes. Note that these differences arise only from differences in $XCO_2^{model}$, and therefore large differences between the $XCH_4^{proxy}$ measurements point towards high uncertainties in the model representations of atmospheric $CO_2$. Figure
5 6 further displays the result of these inversions. We find a mean difference between $XCH_4^{ma}$ and $XCH_4^{ct}$ of -2.36 ppb and a $\sigma$ of 4.55 ppb. This is caused by mean differences between $XCO_2^{ma}$ and $XCO_2^{ct}$ of -0.50 ppm and a $\sigma$ of 0.97 ppm (not shown in the Figure). We find a seasonal variation in the difference with the largest amplitudes of about 10 ppb in the northern tropics. The phasing varies with latitude, with positive values during boreal summer to autumn. The smallest differences are found in the Southern Hemisphere. The bottom panel of Figure 6 shows how this seasonal pattern propagates into the posterior $CH_4$
10 fluxes. The seasonal and latitudinal variation in the $CH_4$ flux difference follows the variation in the $XCH_4^{proxy}$ difference, with an amplitude of 0.5 Tg/month/gridcell. The regions without satellite data coverage, i.e. below 60° S and above 60° N, show smaller differences in the optimized fluxes.

PR-CT and PR-MA yield different $CH_4$ fluxes in Northern Africa and Australia (see Figure 4). We plot these fluxes with the corresponding regional averaged $XCH_4$ values in Figure 7. For Northern Africa, the difference in $XCH_4^{proxy}$ of up to 10
15 ppb around January 2010 gives rise to a difference in the monthly posterior flux of   1 Tg/month. In Australia, $XCH_4^{ma}$ and $XCH_4^{ct}$ are in relatively good agreement with each other, with differences within 2 ppb. However, because the *a priori* emission from this region is very small, the difference in the optimized seasonal cycle of fluxes nevertheless becomes relatively large. In particular PR-MA causes significant deviations from the *a priori*, with decreases in the posterior fluxes during Australian



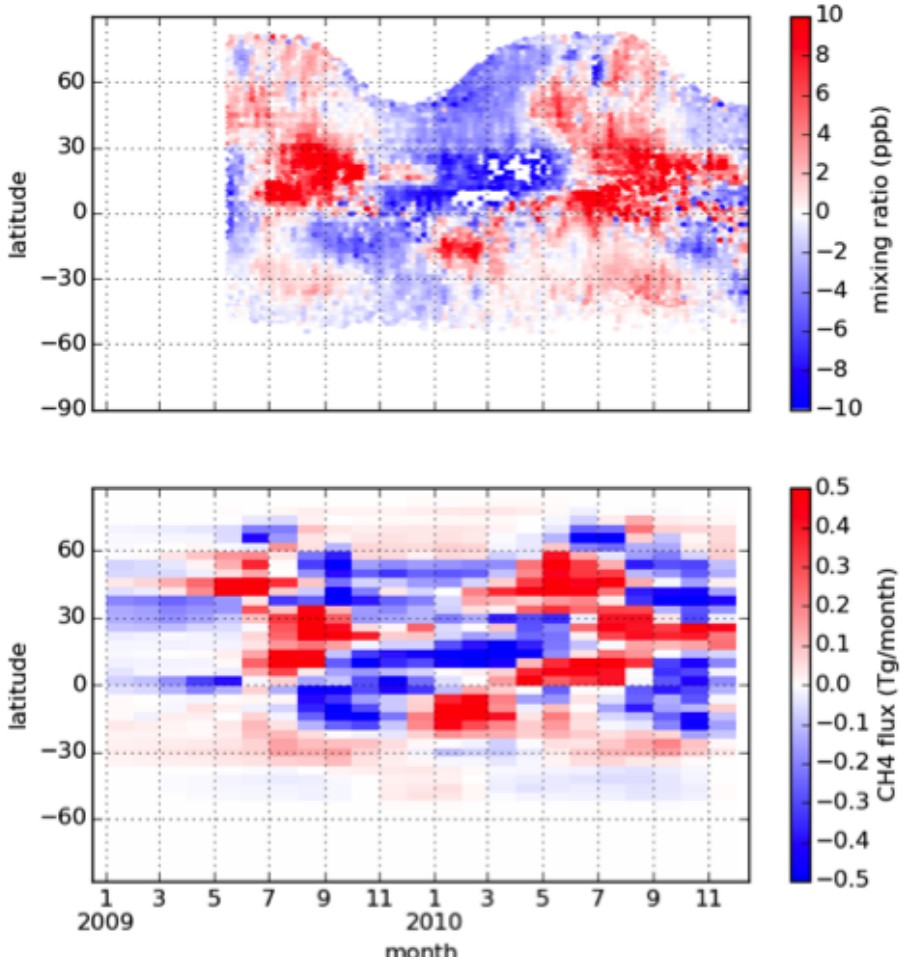

**Figure 6.** Top: Zonally averaged differences in $CH_4$ column mixing ratio between the two $XCH_4^{proxy}$ retrievals ($XCH_4^{ma}$ - $XCH_4^{ct}$). Bottom: Corresponding differences in *a posteriori* $CH_4$ flux between the proxy inversions using these data (PR-MA minus PR-CT).

summer, and large increases during winter. Another reason for these flux adjustments is the limited land area in the Southern Hemisphere that is available for $CH_4$ flux adjustments (over the open ocean the *a priori* flux uncertainties are small limiting their adjustment).

(Detmers et al., 2015) reported an enhanced $CO_2$ sink over Central Australia in the second half of 2010 lasting until 2012, caused by an increase in vegetation due to enhanced precipitation during *La-Nina* conditions. If not properly represented in inversions using surface measurements, this negative $CO_2$ anomaly causes $XCO_2^{model}$ to be overestimated. In that case, the anomaly propagates to the proxy retrievals resulting in overestimation of $XCH_4^{proxy}$, leading to overestimated *a posteriori* $CH_4$





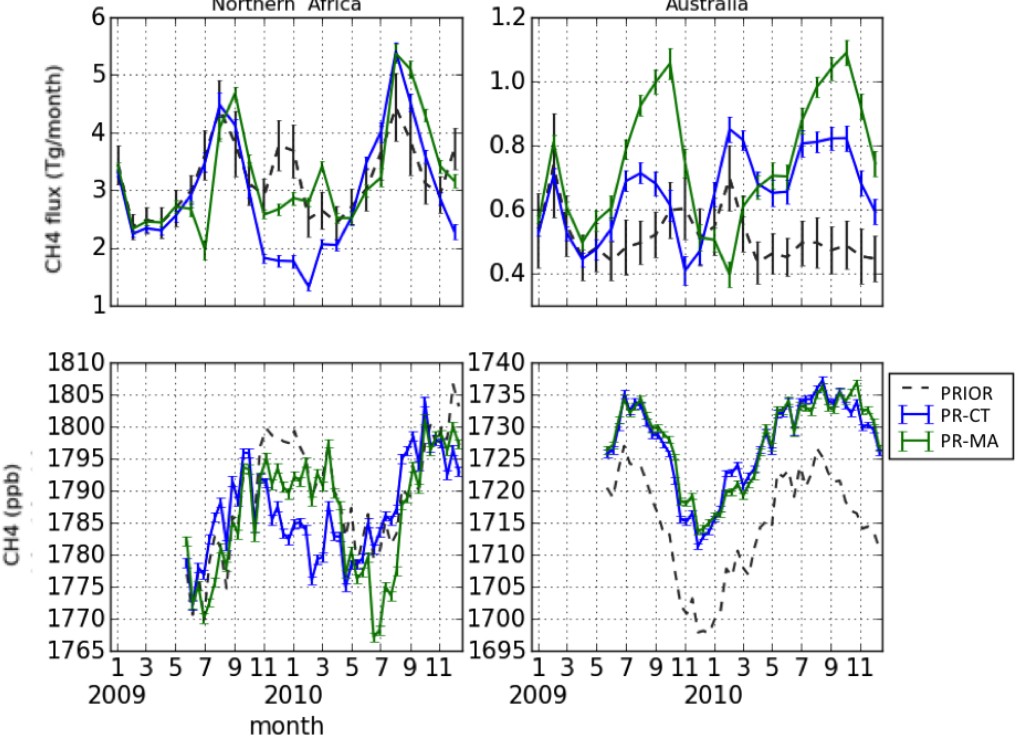

**Figure 7.** The top panels shows the posterior monthly fluxes integrated over TRANSCOM region. Bottom panels show the time series of the mean of $XCH_4^{proxy}$ over Northern Africa and Australia. The dotted line in the bottom panels denotes the mean of *a priori* modeled $XCH_4$ sampled at GOSAT sites.

fluxes. RATIO estimates a significantly stronger sink of $CO_2$ in agreement with (Basu et al., 2013) (see supplementary Figure 2). This results in lower $CH_4$ fluxes in the RATIO inversion (see Figure 4), demonstrating how the RATIO inversion method can avoid shortcomings in the proxy inversions in regions where $CO_2$ is poorly constrained by surface data.

PR-CT and PR-MA have opposite seasonal cycles, which may be due to their $XCO_2^{model}$ components, which are derived using
5  different ecosystem models. Carbontracker uses *a priori* natural fluxes from a CASA simulation driven by actual climatological information, whereas MACC uses only the climatology of natural fluxes. Therefore, the inter-annual variability of the inverted fluxes in MACC is driven by measurements only. Since the surface network does not pose strong constraints on the Australian carbon budget, the differences are driven by the prior fluxes of the two models, which may be more realistic in Carbontracker in this case.

10  **3.2.5  Aircraft Validation**

To further investigate the performance of our inversions, we validate the inversion-optimized $CH_4$ and $CO_2$ mixing ratios against independent aircraft measurements obtained during the projects described in section 2.2. The results of the HIPPO and



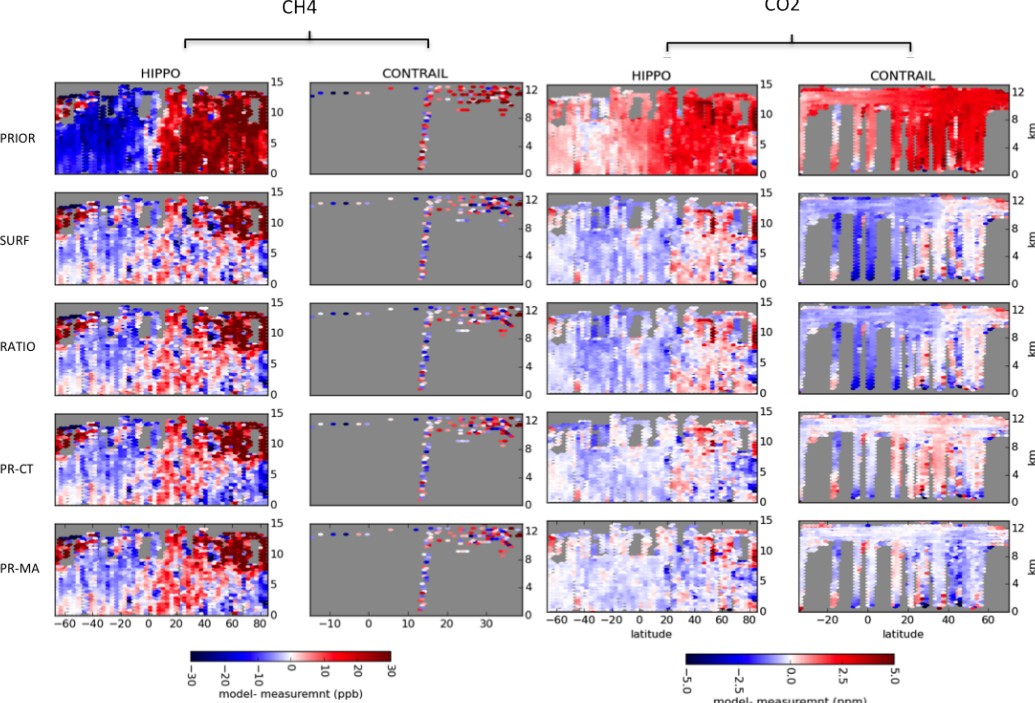

**Figure 8.** Validation of inversion-optimized concentration fields of $CO_2$ and $CH_4$ with air-borne measurements.

CONTRAIL validation are shown in figure 8 and the values for $\kappa$ for $CH_4$ and the root mean square difference (RMSD) for $CO_2$ are given in 9. $\kappa$ values are not calculated for $CO_2$ because we do not have the $CO_2$ model representation errors used in MACC and CarbonTracker. More details on statistics of the validation are provided in Table 1.

    The difference between HIPPO and PRIOR reflects the overestimated north-south gradient that is found using *a priori* $CH_4$
5   fluxes, as already discussed in section 3.2.2. In addition, PRIOR shows a uniform bias of 13.5 ppb. SURF and RATIO correct the north-south gradient and reduce biases to 5.56 and 6.68 ppb, respectively. All the models are performing equally well in terms of $\kappa$. The original MACC and CarbonTracker $CO_2$ fields have RMSD values of 1.08 and 1.09 ppm, respectively, which is lower than the RMSD of RATIO (1.64 ppm) and SURF (1.65 ppm). We suggest that CarbonTracker and MACC have a better representation of $CO_2$ than PR-CT, PR-MA and SURF as they assimilate a larger number of flask measurements sites and also
10  few continuous *in-situ* sites.

    Compared with the large CONTRAIL dataset of $CO_2$ measurements, only a limited number of $CH_4$ measurements are available, mostly over the Pacific Ocean (see Figure 1). We observe the same north-south gradient mismatch with PRIOR as seen in the comparison to HIPPO. PR-CT is able to improve the PRIOR $\kappa$ of 6.99 to 4.56, followed in order of decreasing performance by PR-MA (4.71), SURF (5.33), and RATIO (5.47). The values of $\kappa$ are larger than 1, which points to significant
15  errors in all the inversion results. The RMSD of the different inversions are comparable. The large dataset of CONTRAIL $CO_2$





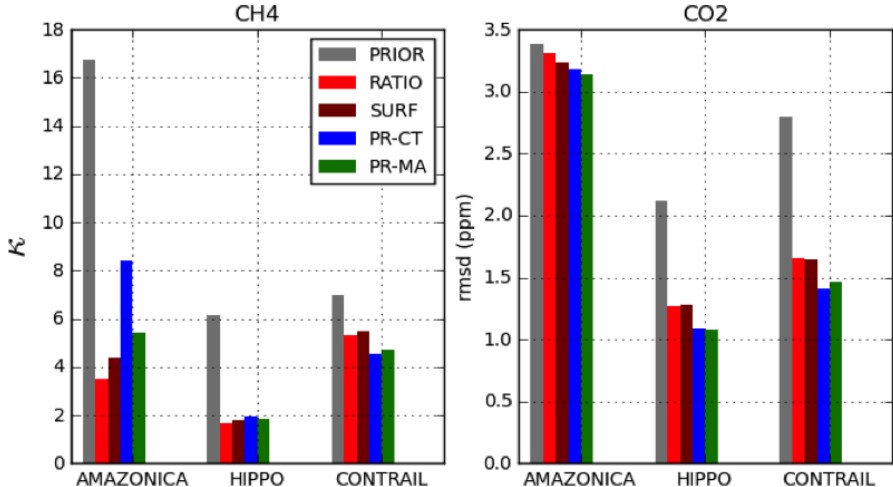

**Figure 9.** Summary of aircraft validation results per project for $CH_4$ (left panel, expressed as $\kappa$) and $CO_2$ (right panel, expressed as RMSD) for the whole inversion time period i.e. from 1/1/2009 to 31/12/2010.

measurements covers a much larger area, including flight tracks to Europe and South East Asia. Our validation shows a mean error of 2.23 ppm in PRIOR. The NOAA and RATIO inversions reduce this bias to -0.43 ppm and -0.41 ppm, respectively. However, similar to the HIPPO validation, MACC (mean bias = -0.2 ppm) and CarbonTracker-derived $CO_2$ (mean bias = 0.11 ppm) fields are in better agreement with the CONTRAIL measurements than the other inversions.

### 3.2.6 Tropical South America

Tropical South America contains the Amazon basin, which is a large reservoir of standing biomass and contains one of the largest wetlands in the world. Therefore, it plays an important role in the annual global budget of both $CO_2$ and $CH_4$. Inversion results for the region have been validated using AMAZONICA measurements (see Supplementary Figure 5 and Supplementary Table 1). Generally, the model results using PRIOR emissions underestimate the measured $CH_4$ mixing ratios (mean offset= -32.02 ppb). All inversions correct this offset, with SURF performing best (mean offset= -14.18 ppb). RATIO closely follows SURF with a mean mismatch of -17.18 ppb. The proxy inversions have a higher mismatch than RATIO and SURF, with means of -20.30 and -24.11 ppb respectively for PR-MA and PR-CT. The $\kappa$ values for the AMAZONICA $CH_4$ measurements (see Figure 9) again show that fluxes from RATIO lead to lower mismatches than those from PR-CT and PR-MA. RATIO predicts this region as a significantly high $CH_4$ source for the first half of 2010 (see Figure 4), and is in good agreement with aircraft measurements.

To check whether this is caused by errors in $XCO_2^{model}$, we perform similar comparisons using AMAZONICA $CO_2$ measurements. We find that the two original models represent $CO_2$ about equally well in terms of RMSD (see Supplementary Table 1). Therefore, the lesser performance of PR-CT and PR-MA for $CH_4$ is not due to a poor representation of the $XCO_2^{model}$ over the



region. This raises the question why RATIO performs better? In sections 3.1, we observe that the error in $CO_2^{model}$ is generally lower than the error in the GOSAT $X_{ratio}$ retrievals (see section 3.1. In proxy inversions, this retrieval error ,which is coming from $X_{ratio}$ (see equation 2), is directly transferred to $CH_4$ fluxes, whereas in RATIO it is distributed over the $CH_4$ and $CO_2$ part of the state vector. The high posterior $CO_2$ flux uncertainties for RATIO in the region support this further (see Figure 5).

Flux maps of the region show that the satellite inversions provide a more spatially resolved adjustment of the $CH_4$ fluxes than SURF (see Supplementary Figure 3). The satellite inversions estimate higher fluxes in the northwest corner of the region near Columbia. Similar increases have been reported in earlier studies assimilating satellite retrieved $XCH_4$ (Monteil et al., 2013; Frankenberg et al., 2006). The spatial pattern of the flux adjustment suggests that the proxy inversions compensate the increase over Columbia by reducing the fluxes in the Amazon Basin, which is less well covered by satellite retrievals due to

frequent cloud cover. This may explain why the proxy inversions end up underestimating the observations inside the Basin. SURF is mainly constrained by the large-scale inter-hemispheric gradient. This leads to a different pattern of flux adjustments, increasing only the fluxes in the southern part of the region while keeping the fluxes in Amazon Basin close to the prior. This solution brings SURF in relatively close agreement with the measurements. RATIO also shows a flux enhancement in Columbia, but at the same time represents the Amazon Basin better than the proxy inversions, likely because of its larger

number of degrees of freedom in modifying regional flux patterns of both $CO_2$ and $CH_4$.

    Gatti et al. (2014) and Vanderlaan et al. (2015) reported an anomalous natural source of $CO_2$ in the region in 2010, also using AMAZONICA aircraft measurements. In this study, RATIO predicts a more enhanced $CO_2$ natural source than the SURF and PRIOR. RATIO (RMSD =3.23 ppm) is also in better agreement in terms of RMSD with AMAZONICA $CO_2$ data than SURF (RMSD=3.31 ppm) and PRIOR (3.38 ppm). This demonstrates, like in the case of Australia, that the RATIO method is capable

of informing us about the $CO_2$ fluxes, from which the $CH_4$ flux estimation benefits also.

### 3.2.7   Temperate Eurasia

  As mentioned in section 3.2.2, SURF leads to a drastic emission reduction in Temperate Eurasia, whereas all satellite inversions show comparatively smaller decreases. Here, we investigate this in further detail by analyzing the inversion-optimized fits to the NOAA measurements at five surface sites located in this region (Figure 10). We find large mismatches between the *a priori*

simulated concentrations and the measurement at these sites, with mean offsets ranging between 29.1 ppb at Mt. Waliguan and 174 ppb at Shangdianzi. All inversions correct for this mismatch by decreasing the regional emissions. Surprisingly enough, the satellite inversions are able to fit the flask measurements even better than SURF, despite smaller corrections to the fluxes. For example, the mean posterior mismatch at Shangdianzi is 24.3 ppb for SURF, and only 7.5 ppb to 9.8 ppb for the satellite inversions. A possible explanation is the double counting of surface data in the satellite inversions, because the satellite data

have been bias corrected using an inversion that was already optimized using surface data. However, the bias correction is only applied as a zonal and annual mean. All inversions show similar reductions in the fluxes from eastern Temperate Eurasia (mostly China) to match the NOAA measurements. However, the satellite inversions tend to compensate for this flux decrease over China by increased fluxes in India and the central part of Temperate Eurasia.



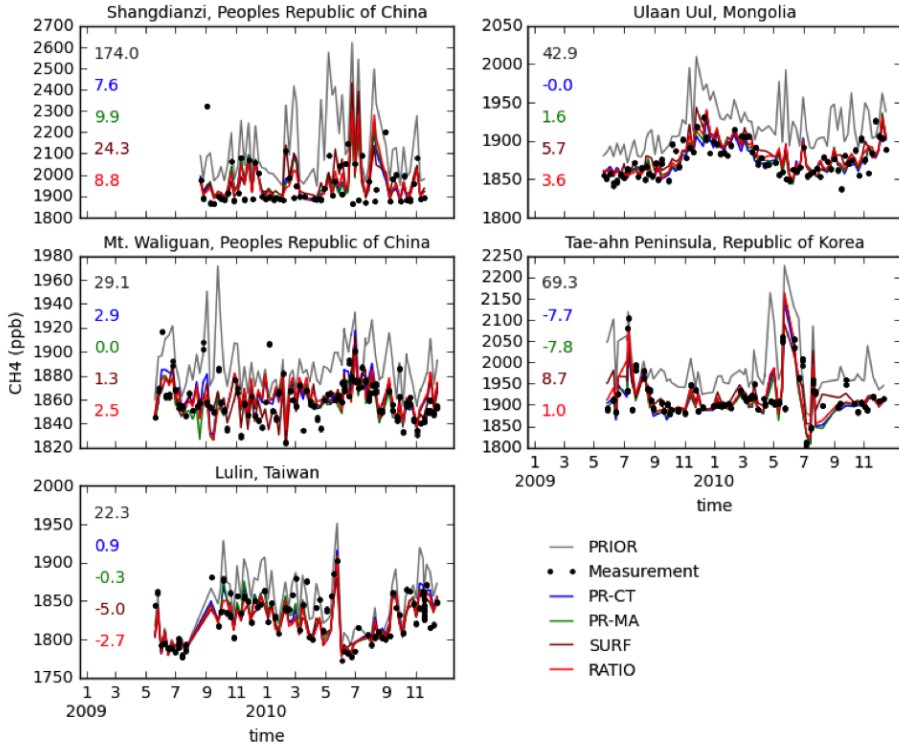

**Figure 10.** Inversion-optimized fits to surface measurement sites in Temperate Eurasia. The numbers in the plots are the mean bias of models with measurements.

## 4 Disscussion

We have demonstrated that the application of the ratio method to GOSAT data yields realistic solutions for $CO_2$ and $CH_4$ fluxes. Its performance is comparable, and may in some regions even be better than the proxy inversion method. This is an important finding because the $X_{ratio}$ retrieval approach provides a useful alternative to the full-physics method in that cloud
5    filtering is less critical. In the case of GOSAT, it increases the number of useful measurements by about a factor of two (Butz et al., 2010; Fraser et al., 2014). At the same time, the RATIO inversion method avoids using the model-derived-$CO_2$ fields as a hard constraint, which is the an important limitation of the proxy method.

    The realistic performance of the ratio method is certainly not a trivial outcome, since it prompts the user for specification of new uncertainties influencing the way in which measurement information is shared between $CH_4$ and $CO_2$. The joint $CO_2$ and
10   $CH_4$ inversion problem has a larger number of degrees of freedom, as a result of which $CH_4$ flux adjustments can compensate for errors in $CO_2$ and vice versa. Assimilating surface measurements helps decoupling $CH_4$ and $CO_2$, which works best in regions that are relatively well covered by the surface network.





In other regions, the method can be improved further by accounting for correlations between *a priori* fluxes of $CH_4$ and $CO_2$. This study does not specify such correlations, which corresponds to the assumption that *a priori* $CO_2$ and $CH_4$ flux uncertainties are independent of each other. Fraser et al. (2014) accounted for *a priori* uncertainty correlations for biomass burning fluxes of $CO_2$ and $CH_4$, based on the available information about emission ratios. Imposing such *a priori* constraints

increased posterior uncertainty reduction compared to other methods for both $CH_4$ and $CO_2$ in some regions.

One problem with the ratio method is the assimilation of $X_{ratio}$ over oceans. The uncertainty of $CH_4$ fluxes over the open oceans is relatively small. As a result, the model-data mismatch over the ocean is mostly accounted for by adjusting the $CO_2$ fluxes, which has a larger *a priori* uncertainty. At the same time, $CO_2$ fluxes over oceans tend to be very sensitive to small and systematic model-data mismatches of a few tenths of a ppm (Basu et al., 2013). Any bias in atmospheric transport, affecting

both $CO_2$ and $CH_4$ is projected on the $CO_2$ fluxes, which may lead to rather unrealistic estimates of the annual $CO_2$ exchange over oceanic regions. Palmer et al. (2006) proposed to account for cross correlations in the model representation error between the components of a dual tracer inversion, which could reduce this problem.

Our surface-only inversion shows a large decrease in the fluxes from Temperature Eurasia. To better understand this, we look at results of other recently published $CH_4$ inversion results. We group the studies into three groups: 1. Studies not using

EDGAR v4.2 as prior, comprising of Houweling et al. (2014); Monteil et al. (2013); Bruhwiler et al. (2014); Fraser et al. (2013); 2. Studies using EDGAR v4.2 but not assimilating the Shangdianzi site, comprising of Alexe et al. (2014); Bergamaschi et al. (2013); 3. Studies using EDGAR v4.2 and assimilating Shangdianzi site comprising of this work and Thompson et al. (2015). The inversions of group 1 do not show a systematic reduction in fluxes of Temperate Eurasia. Inversions of Group 3 tend to reduce the emissions from the region the most, whereas, group 2 reduces emissions by an intermediate amount. This outcome

is partly explained by the EDGAR 4.2 emissions being substantially higher in Temperate Eurasia than previous EDGAR versions, as found also by Bergamaschi et al. (2013). In addition, however, these increased emissions have the largest impact on surface-only inversions assimilating measurements from the Shangdianzi site, possibly due to a nearby hot spot in EDGAR v4.2. The hotspot is located near Jiexiu in the Shanxi province (112E, 37N), and has coal emissions of 10.83 Tg/yr for the year 2010 from a $10 \times 10$ km grid. According to the EDGAR team (G. Meanhout, personal communication), this unrealistically

high local source of $CH_4$ is the consequence of disaggregating large emission from Chinese coal mining using the limited available information on the location of the coal mines. Thompson et al. (2015), the other study in group 3, show a large *a priori* mismatch with a root mean square error of 103 ppb at Shangdianzi. Their inversions reduce an *a priori* East Asian $CH_4$ emission of 82 Tg/y by 23 Tg/y, with large adjustments in the emissions from rice cultivation. Further research is needed to investigate the implications of the shortcoming of EDGAR v4.2. It is noteworthy, however, that when satellite data are

assimilated in these studies, the improved regional coverage reduces the impact of this local disaggregation problem on the estimated regional emissions.



## 5 Conclusions

This study investigated the use of GOSAT-retrieved-$X_{ratio}$ for constraining the surface fluxes of $CO_2$ and $CH_4$. First, we validated the $XCH_4$, $XCO_2$ and $X_{ratio}$ retrievals, as well as the model-derived-$XCO_2$ fields used in the proxy methods, using TCCON measurements. This analysis confirmed that biases in non-scattering $XCH_4$ and $XCO_2$ retrievals cancel out in $X_{ratio}$. $X_{ratio}$ has a larger mean bias than model-derived-$XCO_2$ from CarbonTracker and MACC, suggesting that mostly retrieval biases, rather than $CO_2$ model errors, limit the performance of the proxy method. This is true especially at large temporal and spatial scale. To account for biases in GOSAT-retrieved $X_{ratio}$ a TCCON-derived correction was applied as a function of surface albedo, resulting in a mean adjustment of  -0.74%. An additional correction was applied to $X_{ratio}$, $XCH_4^{ct}$ and $XCH_4^{ma}$ to account for a bias between NOAA-optimized-$CH_4$ fields in TM5 and TCCON observed $XCH_4$, amounting to  -0.76%, -0.80% and 0.59%, respectively.

We optimized monthly $CH_4$ and $CO_2$ fluxes for the year 2009 and 2010 by assimilating GOSAT-retrieved-$X_{ratio}$ data using the TM5-4DVAR inverse modeling system. Additional inversions, assimilating $XCH_4^{proxy}$ and NOAA surface flask measurements were performed in a similar setup for comparison. The posterior uncertainties of the fluxes are calculated with a Monte-Carlo approach.

Overall, the ratio and proxy inversions show similar results for annual $CH_4$ fluxes. Significant seasonal differences in $CH_4$ are found between the two proxy inversions for TRANSCOM regions Northern Africa and Australia, which can be traced back to differences in $XCO_2^{model}$. The $CO_2$ models show a systematic difference in the seasonal cycle of $CO_2$, resulting in a seasonally varying mismatch in the northern tropics. The ratio method has the advantage that it allows adjustment of the $CO_2$ fluxes, whereas the proxy inversions can only account for this mismatch by adjusting $CH_4$. For Australia, the proxy inversions predict an anomalous $CH_4$ increase in the second half of 2010. This difference can be explained by errors in $XCO_2^{model}$, which does not account for the anomalous carbon sink reported by Detmers et al. (2015) for lack of surface measurement coverage. The ratio method has the build-in flexibility needed to attribute the anomaly to $CO_2$ instead of $CH_4$ and is therefore is not affected.

Inversions using satellite data show a better agreement among each other compared to the NOAA-only inversions, which use only surface data. This is true in particular for Temperate Eurasia, where the NOAA-only inversion reduces the annual $CH_4$ flux by as much as 55 Tg/y, relative to an *a priori* flux of 121 Tg/y. This is traced back to a large overestimation of atmospheric $CH_4$ concentration in the prior model at NOAA sites in the region, especially at Shangdianzi, where the prior model overestimates the data by 179 ppb on average. When satellite measurements are assimilated, the $CH_4$ flux reduction for Temperate Eurasia is limited to 21 Tg/y, while accounting for the *a priori* mismatch in Shangdianzi.

We validated the inversion-optimized atmospheric tracer fields, as well as the CarbonTracker and MACC $CO_2$ fields used in the proxy inversions, against three independent aircraft measurement projects. For $CH_4$, the ratio and NOAA-only inversions showed a lower mismatch with HIPPO and AMAZONICA measurements than the two proxy inversions. Further analysis shows that this is not due to a better representation of atmospheric $CO_2$ in the ratio inversion. However, the ratio inversion accounts for inconsistent constraints from $X_{ratio}$ by correcting both $CH_4$ and $CO_2$ fluxes, whereas the proxy inversions can only





attribute such constraints to $CH_4$ fluxes. The ratio inversion predicts an enhanced $CO_2$ natural source in this region during 2010 compared with the NOAA-only and *a priori* model. This is accordance with the findings of Gatti et al. (2014) and Vanderlaan et al. (2015), and is also supported by the AMAZONICA aircraft measurements. Overall, this study shows that the ratio method is capable of informing us about surface fluxes of $CH_4$ and $CO_2$ using satellite measurements, and that it provides a useful

5 alternative for the proxy inversion method.





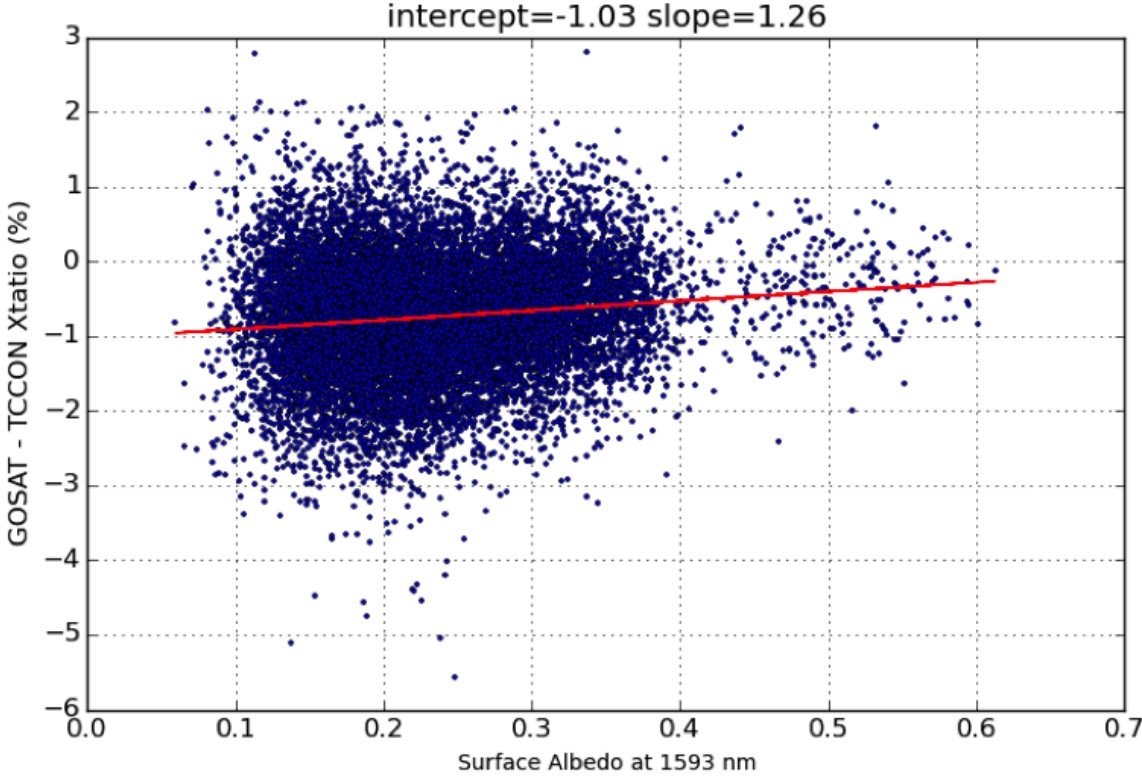

**Figure 11.** Linear regression analysis between GOSAT – TCCON $X_{ratio}$ with Surface albedo at 1593 nm.

**Appendix A: Bias correction**

We apply a two-step correction to reduce the influence of biases in our inversions:

1. **TCCON-based**: Residual biases in $X_{ratio}$ remain that are not accounted for by taking the ratio between $XCH_4^{ns}$ and $XCO_2^{ns}$. The standard bias correction procedure in the RemoTeC $XCH_4^{proxy}$ retrieval assumes a linear dependence on surface albedo (Guerlet et al., 2013). However, this procedure would also correct biases in $XCO_2^{model}$, which are not expected to vary with surface albedo. Therefore, we apply the albedo-based bias correction only to the GOSAT-measured-$X_{ratio}$. To determine the bias correction, we use GOSAT retrievals that are co-located with TCCON measurements, i.e. they are within 5 degrees latitude and longitude and within 2 hours of TCCON measurements. The relationship between surface albedo at 1593 nm and the monthly difference between GOSAT and TCCON is shown in Figure 11 . A global bias correction function is obtained by linear regression, results in a mean adjustment of -0.74% of GOSAT $X_{ratio}$.



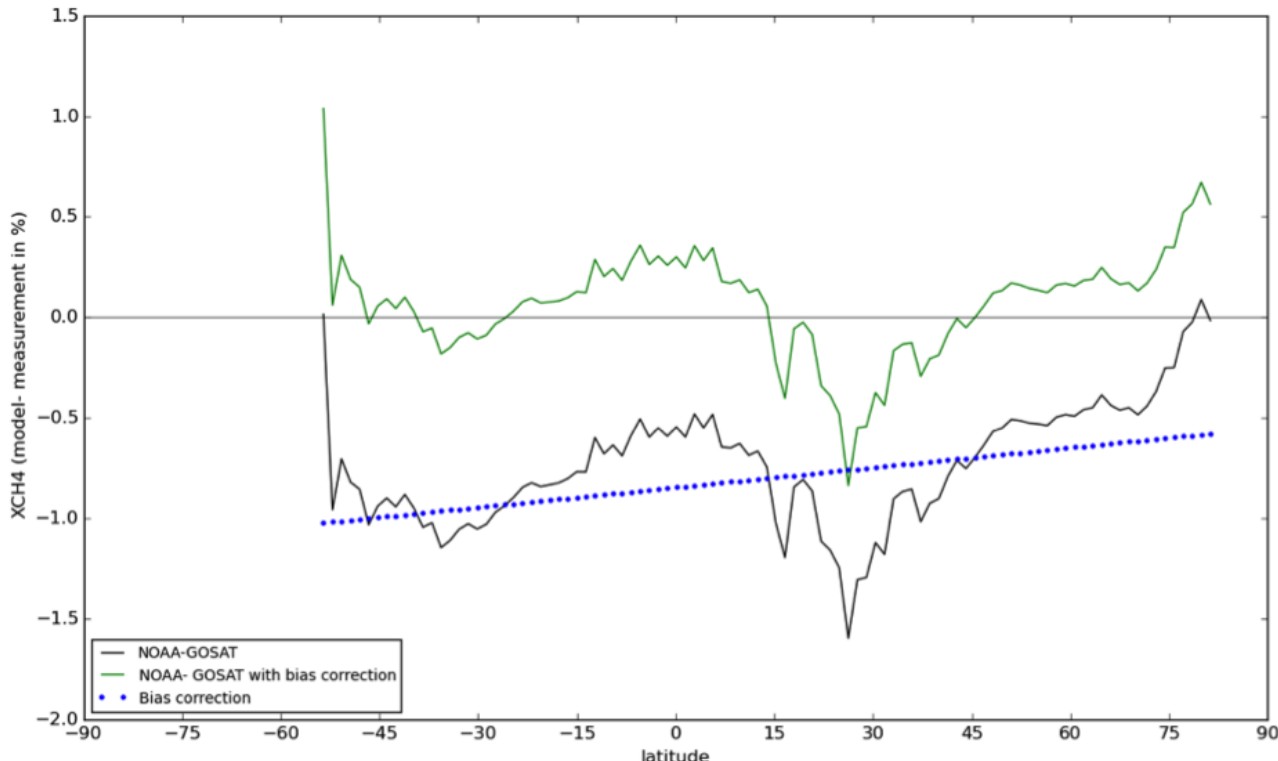

**Figure 12.** NOAA based bias correction applied to XCH$_4$ in the PR-CT inversion

2. **NOAA-based:** A systematic mismatch between the NOAA and GOSAT-optimized TM5 CH4 fields has been discussed in Monteil et al. (2013). The cause of this problem is still unresolved, but may be explained in part by transport model uncertainties in representing XCH$_4$ in the stratosphere. Several other studies have reported similar biases and applied NOAA-based bias corrections, in addition to the TCCON derived retrieval corrections, in order to restore consistency between the observational constraints provided by surface and total column measurements (Alexe et al., 2014; Houweling et al., 2014; Basu et al., 2013). We use a similar procedure for X$_{ratio}$ and XCH$_4^{proxy}$ data by comparing the TCCON-corrected GOSAT retrievals to the NOAA-optimized TM5 model. The mean difference is corrected using a linear function of latitude. This results in a mean adjustment of -0.76 % in X$_{ratio}$, -0.59% in XCH$_4^{ma}$ and -0.80% in XCH$_4^{ct}$ (See Figure 12 and 13)

## Appendix B: Posterior Uncertainty

As discussed in Pandey et al. (2015), the X$_{ratio}$ inversion problem is weakly non-linear and is solved using the quasi–Newtonian optimizer M1QN3. The standard implementation of M1QN3 does not provide an estimate of posterior uncertainty. Therefore,





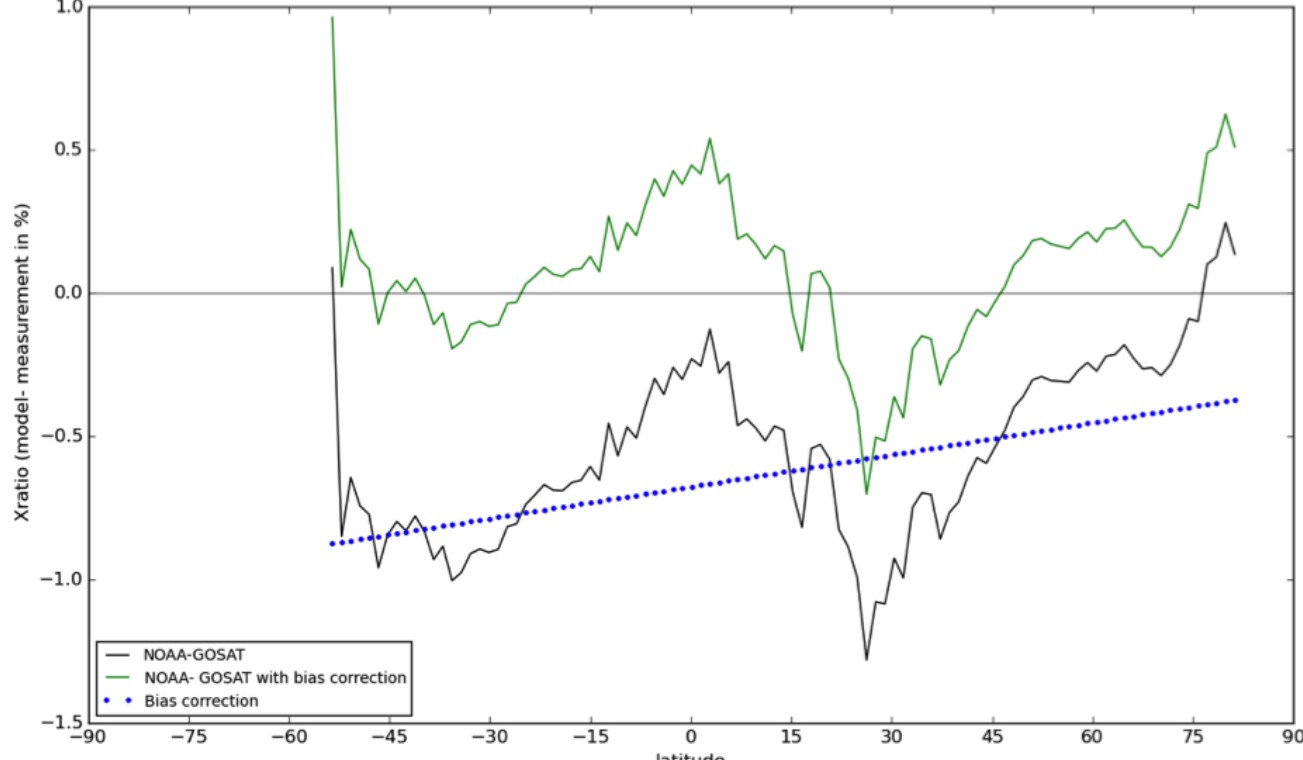

**Figure 13.** NOAA based bias correction applied to $X_{ratio}$ in the RATIO inversion

we use the Monte-Carlo approach as described inChevallier et al. (2007) to calculate posterior flux uncertainties. For the linear SURF and proxy inversions, which use the conjugate gradient optimization method. The posterior flux uncertainties of these inversions are derived using the same approach to keep the comparison between the uncertainties consistent. A sensitivity test has been performed to determine the size of the ensemble needed to properly capture the $1 \sigma$ of the prior fluxes. Figure 14

5     shows the results of this experiment. We choose an ensemble size of 24 for our experiments which gives a $1 \sigma$ estimate with 14.4 % uncertainty.

*Acknowledgements.* This work is supported by the Netherlands Organization for Scientific Research (NWO), project number ALW-GO-AO/11-24. The computations were carried out on the Dutch national supercomputer Cartesius, and we thank SURFSara (www.surfsara.nl) for their support. Access to the GOSAT data was granted through the third GOSAT research announcement jointly issued by JAVA, NIES,

10     and MOE. The funding for AMAZONICA project is provided by NERC and FAPESP. We thank S. C. Wofsy for providing HIPPO data. We thank Debra Wunch and other TCCON PI's for making their measurements available.

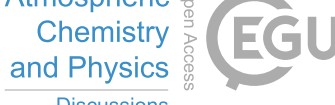

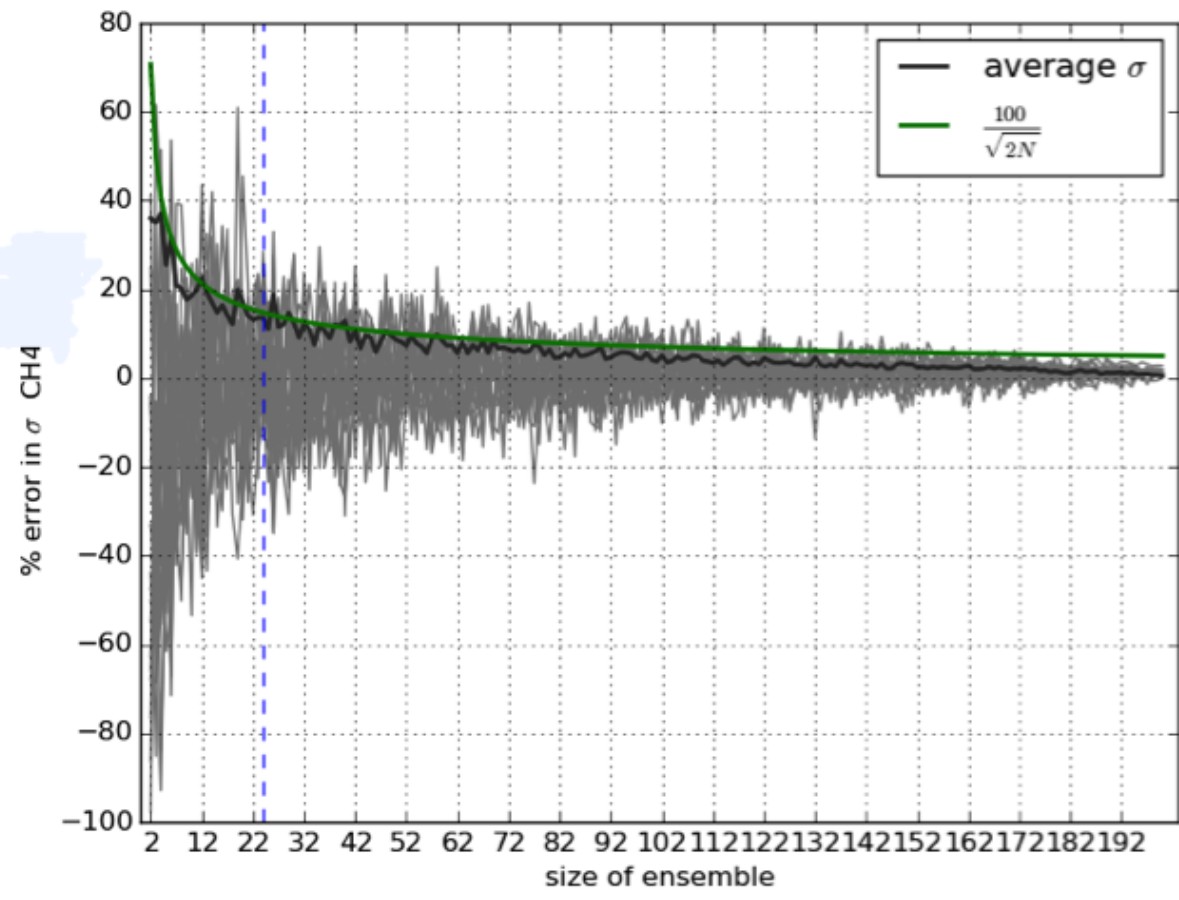

**Figure 14.** The gray lines represent the percentage error of $\sigma$ of ensemble size n from the $\sigma$ of ensemble size of 200 of the *a priori* CH4 flux integrated over TRANSCOM regions. The dark black line represents the average deviation in the grey lines. Green line represent the analytical variation of the uncertainty of $1\sigma$ (Bousserez et al., 2015). A constant difference of approx. 6% between the estimates comes from the finite size of the largest sample (200).



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
