# Peer review of "Inverse modeling of GOSAT-retrieved ratios of total column $CH_4$ and $CO_2$ for 2009 and 2010."

_Atmospheric Chemistry and Physics, 2016_

## Referee Comment (RC1) · Anonymous Referee #1 · 29 Feb 2016

This paper presents an inversion of CH4 and CO2 using GOSAT column retrievals and surface observations. The central theme of the paper is a comparison of flux inversions derived from an assimilation of the XCH4/XCO2 ratio (also constrained by surface observations) to "proxy" or "surface only" retrievals. The paper is suitable for publication in ACP, provided that some of the comments below are addressed.

General comments

- Use of bias correction. As the authors themselves note (P18, line 29), there appears to be a double counting of the surface observations in the satellite inversions, because a "bias correction" has been applied, based on previous satellite-only and surface-only inversions. In addition to this double-counting, I suggest that there are two problems with this approach: a) any discrepancy between these two inversions is likely to be an

indicator of systematic model errors, which are likely to result in a relatively complex "offset" between two such inversions (indeed, this appears to be indicated in Figure 12). Therefore, the use of a linear "correction" would leave out some potentially important features un-accounted for; b) any uncertainty associated with this correction is not propagated through the inversion. Instead of imposing this correction as a hard constraint, why not include it in the inversions? Point a) above could also be addressed by disaggregating this potential error into more than two components (i.e. an intercept and a gradient).

- Uncertainty quantification. Given the rather extensive discussion of potential biases and uncertainties associated with the retrievals and the model, the uncertainties derived in Figures 3, 4, 5 and 7 seem very optimistic to me. I think the paper would benefit from a much broader discussion about any limitations in the uncertainty quantification methods employed in this paper (preferably with reference to other methods that have been used in the literature). This should also include a discussion of the choices made about the a priori uncertainties. For example, it appears that a choice of 50% uncertainty for each grid cell for the CH4 prior was used with a temporal correlation of 3 months and length scale of 500km. Why were these figures chosen? What might be the influence on the inversion of choices of this nature? When aggregated together, it appears that this uncertainty leads to a prior uncertainty on continental scales of about 5-10% (Figure 3). Does this seem realistic? It seems small to me, especially since it is apparently inconsistent with the outcome of the inversion for several regions. There are several systematic factors that could also strongly influence the outcome of the inversion (e.g. convection, OH uncertainty), which should be discussed.

Minor comments

- P2 L5: "still consistent results are obtained". Consistent with respect to what?

- P2 L10: "original information"? I'm not sure what this means.

- P4, Section 2: This could do with a brief overview of the chemical transport model

setup (fluxes, OH fields, dynamics, etc).

- P5 L4:7. Are these choices largely subjective? Reasons should be given, and a discussion of the implications. See general comment above.

- P5 L14: Missing full stop before The RemoTeC

- P5 L18: referred to as

- P5 L25: Use of "additional" here... additional to what (I presume it means in addition to the TCCON bias correction, which is described in the section below).

- P8 L1: Use of uncorrelated observation/model errors. Is it realistic to assume that the observation and model representation errors are uncorrelated in space and time? What might be the implications of this choice?

- P8 L18: Re-word "relatively less errors" is not grammatically correct.

- P11 L9: "do not show an important seasonal dependence". This needs re-wording. I don't understand what an "important seasonal dependence" means.

- P11 L18: "information that is used" (remove comma).

- P17 L18: "poorer" rather than "lesser"

- P21 L4: Isn't it fairer to say that they "largely cancel out"? The cancellation is not 100%.

- P23 L8: Isn't 5x5 degrees quite a large area? Can this analysis be carried out over smaller scales?

---

## Referee Comment (RC2) · Anonymous Referee #2 · 24 Mar 2016

General comments.

The study by S. Pandey and coauthors reports inverse modeling experiments testing use of the GOSAT-retrieved ratio of methane and carbon dioxide column average concentrations for inverse modeling of both CO2 and CH4 surface fluxes. The manuscript does present new results of considerable interest, and can be accepted with a minor revision. Technical correction and proofreading is needed as there are many mistypes.

Detailed comments.

The ratio in hand is composed of 2 variables that vary very little around mean values. Linear expansion around mean state will transform the difference to a linear combination of XCO2 and XCH4, that is d(XCH4/XCO2)= (dXCH4-dXCO2*XCH4/XCO2)/XCO2.

Given the ratio of column mean concentration around 400/1.7 ppm/ppm, XCH4 gets about 200 times higher weight in the linear combination of the two. Mysteriously, the ratio of the XCO2 and XCH4 errors is about same order (2/0.012 ppm/ppm), so the correlated parts of the errors are largely cancelled in ratio. On the other hand, ratio of surface fluxes is in order of 10/0.3=30 for anthropogenic (according to EDGAR data), and 9/0.2=18 GtC/GtC for natural fluxes (growing season net flux by Randerson et al. 1996; wetlands in Melton et al. 2013). Thus, we have ample imbalance of 6-10 times in favor of methane in terms of signal to noise ratio for sensitivity of XCH4 to XCO2 ratio to surface fluxes. Accordingly, use of a retrieved ratio for CH4 flux inversion is better justified than application for CO2 flux inversion. That makes results of this study interesting to look in. In particular, latitude dependent XCH4 bias contributed by combination of model (stratosphere) and retrieval biases comes in place of reduced aerosol and cloud effects. It would be useful to add discussion on the contribution of the methane XCH4 biases to CO2 inversion constrained by XCH4/XCO2 ratio.

Technical corrections.

Page 01- Line 03 Putting here "biased" instead of "heavily biased" would suffice, referring to current state of retrievals.

02-17 and 02-25 Lists of papers are similar, likely to present same information twice, better to put some distinction. Adding Deng et al ACP 2014 and Maksyutov et al ACP 2013 may be useful for completeness.

02-31 "two types of retrieval methods" can be used in place of "two retrieval methods"

03-21 (Fraser et al., 2014) -> Fraser et al., (2014)

04-27 As- sessment -> Assessment

04-26 right spell should be v.4.2 FT2010

05-01 onJacobson -> on Jacobson

05-12 adding reference to Remotec (Butz?) would help here.

08-04 GOSAST -> GOSAT.

08-20 Should ppm/ppm be used in place of ppb/ppm?

09-01 Units of table 1 need more explanation. Text says it is percentage difference weighted with GOSAT+TCCON error, the value doesn't look like percentage.

25-01 inChevallier -> in Chevallier

28-31 In the reference list initials like A. are appearing as a. in multiple locations.

References.

Randerson, J. T. and coauthors, Substrate limitations for heterotrophs: Implications for models that estimate the seasonal cycle of atmospheric CO2, Global Biogeochem. Cycles, 10(4), 585–602, doi:10.1029/96GB01981, 1996.

Melton, J. R., and coauthors, Present state of global wetland extent and wetland methane modelling: conclusions from a model inter-comparison project (WETCHIMP), Biogeosciences, 10, 753-788, doi:10.5194/bg-10-753-2013, 2013.

---

## Author Comment (AC1) · 4 Apr 2016

We thank the referee for his/her useful comments. We have included the referee's comments (*italics*) and comments specific replies (AC) below. The corresponding changes made in the manuscript are written in blue below the ACs.

**Anonymous Referee 1**

*This paper presents an inversion of CH4 and CO2 using GOSAT column retrievals and surface observations. The central theme of the paper is a comparison of flux inversions derived from an assimilation of the XCH4/XCO2 ratio (also constrained by surface observations) to "proxy" or "surface only" retrievals. The paper is suitable for publication in ACP, provided that some of the comments below are addressed.*

[Figure]

**General Comments:**

*- Use of bias correction. As the authors themselves note (P18, line 29), there appears to be a double counting of the surface observations in the satellite inversions, because a "bias correction" has been applied, based on previous satellite-only and surface-only inversions. In addition to this double-counting, I suggest that there are two problems with this approach:*

*a) any discrepancy between these two inversions is likely to be indicator of systematic model errors, which are likely to result in a relatively complex "offset" between two such inversions (indeed, this appears to be indicated in Figure 12). Therefore, the use of a linear "correction" would leave out some potentially important features un-accounted for;*

AC: Previous analyses have shown that systematic errors are likely caused by a combination of errors in the transport model and the GOSAT retrievals (Monteil et al., 2013). It is better to take them out to avoid that these errors impact the fluxes. We agree with the reviewer that our approach of using a linear bias correction may leave some important features unaccounted. At the same time, using a higher order bias correction can also remove some of the information content of the GOSAT measurements. Linear bias correction is not an unsuitable choice keeping in mind the trade-off.

*b) any uncertainty associated with this correction is not propagated through the inversion. Instead of imposing this correction as a hard constraint, why not include it in the inversions? Point a) above could also be addressed by disaggregating this potential error into more than two components (i.e. an intercept and a gradient).*

AC: Optimizing the corrections further in the inversion will lead to a compromise between fluxes and bias adjustments. We do not trust this component of the flux adjustment, as it might be different for the proxy and ratio inversions that we want to compare. It would add further complexity to our analysis.

*- Uncertainty quantification. Given the rather extensive discussion of potential biases and uncertainties associated with the retrievals and the model, the uncertainties derived in Figures 3, 4, 5 and 7 seem very optimistic to me. I think the paper would benefit from a much broader discussion about any limitations in the uncertainty quantification methods employed in this paper (preferably with reference to other methods that have been used in the literature). This should also include a discussion of the choices made about the a priori uncertainties. For example, it appears that a choice of 50% uncertainty for each grid cell for the CH4 prior was used with a temporal correlation of 3 months and length scale of 500 km. Why were these figures chosen? What might be the influence on the inversion of choices of this nature? When aggregated together, it appears that this uncertainty leads to a prior uncertainty on continental scales of about 5-10% (Figure 3). Does this seem realistic? It seems small to me, especially since it is apparently inconsistent with the outcome of the inversion for several regions. There are several systematic factors that could also strongly influence the outcome of the inversion (e.g. convection, OH uncertainty), which should be discussed.*

AC: It is hard to guess the true uncertainty of the prior fluxes. We take 50% as $1\sigma$ in each grid. If the min/max values are at 95% confidence (i.e. $2\sigma$), our fluxes range between 0 and twice the mean. The correlation lengths help to transfer the uncertainties to larger scales. Earlier studies like Fraser et al., (2013) have also used similar (5-10%) prior uncertainty on continental scales.

The balance between adjustments made to CH4 and CO2 fluxes in the RATIO inversion is also an important factor in our setup. With our present values, the Xratio measurements are twice sensitive to $1\sigma$ changes in CH4 than CO2. As the inversion adjusts the fluxes with respect to the square of model–observation mismatches in the Bayesian framework, CH4 fluxes are adjusted $2^2 = 4$ times more than CO2 fluxes (please see our reply to the 1st comment of 2nd reviewer). If we increase the CH4 prior uncertainty, this number will become even higher making the CO2 fluxes too restricted.

We have made the following update to our manuscript (Discussion Section) to address

the optimistic estimates of posterior uncertainties:

"It is noteworthy that the inversions are run assuming uncorrelated measurements and a perfect transport. Also, as we are not optimizing the atmospheric sink of CH4, all the information from its budget is used to constrain the surface fluxes. Hence, the estimates of posterior uncertainties tend to be optimistic in this study. The $\chi^2$ statistic indicates whether the assumed measurement and prior errors are statistically consistent (Meirink et al., 2008). We find $\chi^2/n_s$ = 0.93 for RATIO, 0.96 for PR-CT, 0.93 for PR-LM and 1.14 for SURF in the CH4 inversions ($n_s$ is the number of observations assimilated in the inversion). This shows that we are not drastically underestimating the prior uncertainties in our CH4 inversions."

**Minor Comment**:

*- P2 L5: "still consistent results are obtained". Consistent with respect to what?*

"still consistent results are obtained with respect to other CH4 inversions."

*- P2 L10: "original information"? I'm not sure what this means.*

"Atmospheric measurements of GHGs can provide information about their atmospheric budget. Inverse modeling methods, also known as top-down approaches, have been developed to make use of this information to obtain improved estimates of surface fluxes"

*- P4, Section 2: This could do with a brief overview of the chemical transport model setup (fluxes, OH fields, dynamics, etc).*

AC: we have added the following to our manuscript (in method section):

"We use the TM5-4DVAR inversion modeling system. It is comprised of the Tracer Transport Model version 5 (TM5, Krol et al., 2005) coupled to a variational data assimilation system (4DVAR, Meirink et al., 2008). TM5 simulates the spatiotemporal distribution of a tracer in the atmosphere for a given set of fluxes. In this study, TM5 is

run at a 6×4 degree horizontal resolution and 25 vertical hybrid sigma pressure levels from the surface to the top of the atmosphere. The meteorological fields for this offline model are taken from the European Centre for Medium-Range Weather Forecasts (ECMWF) ERA-interim reanalysis [Dee et al., 2011]."

*- P5 L4:7. Are these choices largely subjective? Reasons should be given, and a discussion of the implications. See general comment above.*

AC: Please refer to our reply to the general comment above.

*- P5 L14: Missing full stop before The RemoTeC*

*- P5 L18: referred to as*

*- P5 L25: Use of "additional" here. . . additional to what (I presume it means in addition to the TCCON bias correction, which is described in the section below).*

*- P8 L1: Use of uncorrelated observation/model errors. Is it realistic to assume that the observation and model representation errors are uncorrelated in space and time? What might be the implications of this choice?*

AC: A Bayesian inverse model should in principle address the correlation of observations and weight them properly. The observations are assimilated in the inversion by comparing them with model-simulated mixing ratios. Therefore, errors in the model can also create a correlation. In practice, correlations are often ignored, both because they are difficult to quantify, and properly taking them into account slows down the inversion systems by a large extent. A prominent illustration of this is given by the numerical weather prediction systems since most of them assume uncorrelated observation errors (but correlated prior errors). Inverses modeling studies of CH4 don't take them into account directly (for example, Alexe et al., 2014, Houweling et al., 2014, Monteil et al., 2013). Studies assimilating SCIAMACHY measurements implemented the binning method to reduce the impact of clustered measurements of SCIAMACHY (Houweling et al., 2014, Monteil et al., 2013). However, this is not so critical for GOSAT, as the

number of soundings is lesser in amount. Also, the main goal of this study is to compare the results of different inversion methods in a consistent setup. All the inversions are done assuming uncorrelated observation/model errors, so this should not affect our results drastically.

*- P8 L18: Re-word "relatively less errors" is not grammatically correct.*

*- P11 L9: "do not show an important seasonal dependence". This needs re-wording. I don't understand what an "important seasonal dependence" means.*

*- P11 L18: "information that is used" (remove comma).*

*- P17 L18: "poorer" rather than "lesser"*

*- P21 L4: Isn't it fairer to say that they "largely cancel out"? The cancellation is not 100%.*

*- P23 L8: Isn't 5x5 degrees quite a large area? Can this analysis be carried out over smaller scales?*

AC: 5x5 degrees area provides us with a sufficient number of GOSAT retrieval around the TCCON sites.

AC: All other minor comments are addressed in the revised manuscript.

**References:**

Alexe, M., Bergamaschi, P., Segers, A., Detmers, R., Butz, A., Hasekamp, O., Guerlet, S., Parker, R., Boesch, H., Frankenberg, C., Scheepmaker, R. A., Dlugokencky, E., Sweeney, C., Wofsy, S. C., and Kort, E. A.: Inverse modeling of CH4 emissions for 2010–2011 using different satellite retrieval products from GOSAT and SCIAMACHY, Atmospheric Chemistry and Physics Discussions, 14, 11 493–11 539, doi:10.5194/acpd-14-11493-2014.

Dee, D. P., Uppala, S. M., Simmons, A. J., Berrisford, P., Poli, P., Kobayashi, S., Andrae,

U., Balmaseda, M. A., Balsamo, G., Bauer, P., Bechtold, P., Beljaars, A. C. M., van de Berg, L., Bid- lot, J., Bormann, N., Delsol, C., Dragani, R., Fuentes, M., Geer, A. J., Haimberger, L., Healy, S. B., Hersbach, H., Holm, E. V., Isaksen, L., Kållberg, P., Köhler, M., Matricardi, M., McNally, A. P., Monge-Sanz, B. M., Morcrette, J.-J., Park, B.-K., Peubey, C., de Rosnay, P., Tavolato, C., Thépaut, J.-N., and Vitart, F.: The ERA-Interim reanalysis: configuration and performance of the data assimilation system, Q. J. Roy. Meteor. Soc., 137, 553–597, doi:10.1002/qj.828, 2011.

Fraser, A., Palmer, P. I., Feng, L., Boesch, H., Cogan, A., Parker, R., Dlugokencky, E. J., Fraser, P. J., Krummel, P. B., Langen- felds, R. L., O'Doherty, S., Prinn, R. G., Steele, L. P., van der Schoot, M., and Weiss, R. F.: Estimating regional methane sur- face fluxes: the relative importance of surface and GOSAT mole fraction measurements, Atmos. Chem. Phys., 13, 5697–5713, doi:10.5194/acp-13-5697-2013, 2013.

Houweling, S., Krol, M., Bergamaschi, P., Frankenberg, C., Dlugokencky, E. J., Morino, I., Notholt, J., Sherlock, V., Wunch, D., Beck, V., Gerbig, C., Chen, H., Kort, E. a., Röckmann, T., and Aben, I.: A multi-year methane inversion using SCIAMACHY, accounting for systematic errors using TCCON measurements, Atmospheric Chemistry and Physics, 14, 3991–4012, doi:10.5194/acp-14-3991-2014.

Krol, M., Houweling, S., Bregman, B., van den Broek, M., Segers, A., van Velthoven, P., Peters,W., Dentener, F., and Bergamaschi, P.: The two-way nested global chemistry-transport zoom model TM5: algorithm and applications, Atmos. Chem. Phys., 5, 417–432, doi:10.5194/acp-5-417-2005, 2005.

Meirink, J. F., Bergamaschi, P., and Krol, M. C.: Technical Note: Four-dimensional variational data assimilation for inverse modelling of atmospheric methane emissions: method and comparison with synthesis inversion, doi:10.5194/acpd-8-12023-2008, 2008.

Monteil, G., Houweling, S., Butz, A., Guerlet, S., Schepers, D., Hasekamp, O., Frankenberg, C., Scheepmaker, R., Aben, I., and Röckmann, T.: Comparison of CH4 inversions

based on 15 months of GOSAT and SCIAMACHY observations, Journal of Geophysical Research: Atmospheres, 118, 11,807–11,823, doi:10.1002/2013JD019760.

---

## Author Comment (AC2) · 4 Apr 2016

We thank the referee for his/her useful comments. We have included the referee's comments (*italics*) and comments specific replies (AC) below. The corresponding changes made in the manuscript are written in blue below the ACs.

**Anonymous Referee 2**

*The study by S. Pandey and coauthors reports inverse modeling experiments testing use of the GOSAT-retrieved ratio of methane and carbon dioxide column average concentrations for inverse modeling of both CO2 and CH4 surface fluxes. The manuscript does present new results of considerable interest, and can be accepted with a minor revision. Technical correction and proofreading is needed as there are many mistypes.*
**Detailed Comments:**

*The ratio in hand is composed of 2 variables that vary very little around mean values. Linear expansion around mean state will transform the difference to a linear combination of XCO2 and XCH4, that is d(XCH4/XCO2)= (dXCH4-dXCO2\*XCH4/XCO2)/XCO2. Given the ratio of column mean concentration around 400/1.7 ppm/ppm, XCH4 gets about 200 times higher weight in the linear combination of the two. Mysteriously, the ratio of the XCO2 and XCH4 errors is about same order (2/0.012 ppm/ppm), so the correlated parts of the errors are largely cancelled in ratio. On the other hand, ratio of surface fluxes is in order of 10/0.3=30 for anthropogenic (according to EDGAR data), and 9/0.2=18 GtC/GtC for natural fluxes (growing season net flux by Randerson et al. 1996; wetlands in Melton et al. 2013). Thus, we have ample imbalance of 6-10 times in favor of methane in terms of signal to noise ratio for sensitivity of XCH4 to XCO2 ratio to surface fluxes. Accordingly, use of a retrieved ratio for CH4 flux inversion is better justified than application for CO2 flux inversion. That makes results of this study interesting to look in. In particular, latitude dependent XCH4 bias contributed by combination of model (stratosphere) and retrieval biases comes in place of reduced aerosol and cloud effects. It would be useful to add discussion on the contribution of the methane XCH4 biases to CO2 inversion constrained by XCH4/XCO2 ratio.*

AC: We agree with the reviewer, however, there are some limitations to the calculation done above. In our inversions, we do not optimize anthropogenic $CO_2$ fluxes. Also, the fluxes are weighted with their respective error in the cost function. We calculate the same number by adding a $1\sigma$ perturbation of global $CH_4$ ($\approx$15Tg/yr) and $CO_2$ (1.2GtC/yr) fluxes in the atmosphere. This will have the corresponding change in the mixing ratio of the tracers in the atmosphere. The change in $CH_4$ mixing ratio will be (1800 ppb x (15 Tg.yr-1/5000 Tg))= 5.4 ppb.yr $^{-1}$ = 0.3%.yr$^{-1}$ . For CO2, it will be (400ppm x 1.30 PgC.yr$^{-1}$/860 PgC)= 0.6 ppm.yr$^{-1}$ = 0.15 %.yr$^{-1}$. So the Xratio will be impacted 0.3/0.15 $\approx$ 2 times more due to $CH_4$ than $CO_2$. As the inversion is adjusted

with the square of the observations, CH4 fluxes will be adjusted $2^2 = 4$ times more than CO2 fluxes. However, this number can be different on regional scales. As CO2 and CH4 surface data are assimilated also and both of them receive approximately equal weight in the inversion, the ratio may be lower. We have added the following to the revised manuscript:

"The signal from Xratio can be up to $\approx$ 4 times more sensitive to adjustment of CH4 fluxes than CO2 fluxes in our inversion setup. In such case, the surface observations of CO2 become the primary constraint on the CO2 fluxes. This can be further verified by looking at Supplementary figure 2, in which the RATIO and SURF inversion show very good agreement. It should be also noted that latitude dependent XCH4 bias contributed by the combination of transport model and retrieval biases become more important in a Xratio inversion while errors reduce due to aerosol and cloud scattering."

**Technical corrections:**

*Page 01- Line 03 Putting here "biased" instead of "heavily biased" would suffice, referring to current state of retrievals.*

*02-17 and 02-25 Lists of papers are similar, likely to present same information twice, better to put some distinction. Adding Deng et al ACP 2014 and Maksyutov et al ACP 2013 may be useful for completeness.*

*02-31 "two types of retrieval methods" can be used in place of "two retrieval methods"*

*03-21 (Fraser et al., 2014) -> Fraser et al., (2014)*

*04-27 As- sessment -> Assessment*

*04-26 right spell should be v.4.2 FT2010*

*05-01 onJacobson -> on Jacobson*

*05-12 adding reference to Remotec (Butz?) would help here.*

[Figure]

*08-04 GOSAST -> GOSAT.*

*08-20 Should ppm/ppm be used in place of ppb/ppm?*

*09-01 Units of table 1 need more explanation. Text says it is percentage difference weighted with GOSAT+TCCON error, the value doesn't look like percentage.*

*25-01 inChevallier -> in Chevallier*

*28-31 In the reference list initials like A. are appearing as a. in multiple locations.* AC: All technical corrections are addressed in the revised manuscript.

References. Randerson, J. T. and coauthors, Substrate limitations for heterotrophs: Implications for models that estimate the seasonal cycle of atmospheric CO2, Global Biogeochem. Cycles, 10(4), 585–602, doi:10.1029/96GB01981, 1996.

Melton, J. R., and coauthors, Present state of global wetland extent and wetland methane modelling: conclusions from a model inter-comparison project (WETCHIMP), Biogeosciences, 10, 753-788, doi:10.5194/bg-10-753-2013, 2013.

---

## Author Response (AR1)

**1 Reply to Referees**

**1.1 Referee 1**

We thank the referee for his/her useful comments. We have included the referee's comments (*italics*) and comments specific replies (AC) below. The corresponding changes made in the manuscript are written in blue below the ACs.

*This paper presents an inversion of CH4 and CO2 using GOSAT column retrievals and surface observations. The central theme of the paper is a comparison of flux inversions derived from an assimilation of the XCH4/XCO2 ratio (also constrained by surface observations) to proxy or surface only retrievals. The paper is suitable for publication in ACP, provided that some of the comments below are addressed.*

**General Comments:**
*- Use of bias correction. As the authors themselves note (P18, line 29), there appears to be a double counting of the surface observations in the satellite inversions, because a bias correction has been applied, based on previous satellite-only and surface-only inversions. In addition to this double-counting, I suggest that there are two problems with this approach:*

*a) any discrepancy between these two inversions is likely to be indicator of systematic model errors, which are likely to result in a relatively complex offset between two such inversions (indeed, this appears to be indicated in Figure 12). Therefore, the use of a linear correction would leave out some potentially important features un-accounted for;*

AC: Previous analyses have shown that systematic errors are likely caused by a combination of errors in the transport model and the GOSAT retrievals (Monteil et al., 2013). It is better to take them out to avoid that these errors impact the fluxes. We agree with the reviewer that our approach of using a linear bias correction may leave some important features unaccounted. At the same time, using a higher order bias correction can also remove some of the information content of the GOSAT measurements. Linear bias correction is not an unsuitable choice keeping in mind the trade-off.

*b) any uncertainty associated with this correction is not propagated through the inversion. Instead of imposing this correction as a hard constraint, why not include it in the inversions? Point a) above could also be addressed by disaggregating this potential error into more than two components (i.e. an intercept and a gradient).*

AC: Optimizing the corrections further in the inversion will lead to a compromise between fluxes and bias adjustments. We do not trust this component of the flux adjustment, as it might be different for the proxy and

ratio inversions that we want to compare. It would add further complexity to our analysis.

*- Uncertainty quantification. Given the rather extensive discussion of potential biases and uncertainties associated with the retrievals and the model, the uncertainties derived in Figures 3, 4, 5 and 7 seem very optimistic to me. I think the paper would benefit from a much broader discussion about any limitations in the uncertainty quantification methods employed in this paper (preferably with reference to other methods that have been used in the literature). This should also include a discussion of the choices made about the a priori uncertainties. For example, it appears that a choice of 50% uncertainty for each grid cell for the CH4 prior was used with a temporal correlation of 3 months and length scale of 500 km. Why were these figures chosen? What might be the influence on the inversion of choices of this nature? When aggregated together, it appears that this uncertainty leads to a prior uncertainty on continental scales of about 5-10% (Figure 3). Does this seem realistic? It seems small to me, especially since it is apparently inconsistent with the outcome of the inversion for several regions. There are several systematic factors that could also strongly influence the outcome of the inversion (e.g. convection, OH uncertainty), which should be discussed.*

AC: It is hard to guess the true uncertainty of the prior fluxes. We take 50% as $1\sigma$ in each grid. If the min/max values are at 95% confidence (i.e. $2\sigma$), our fluxes range between 0 and twice the mean. The correlation lengths help to transfer the uncertainties to larger scales. Earlier studies like Fraser et al., (2013) have also used similar (5-10%) prior uncertainty on continental scales.

The balance between adjustments made to CH4 and CO2 fluxes in the RATIO inversion is also an important factor in our setup. With our present values, the Xratio measurements are twice sensitive to $1\sigma$ changes in CH4 than CO2. As the inversion adjusts the fluxes with respect to the square of modelobservation mismatches in the Bayesian framework, CH4 fluxes are adjusted $2^2 = 4$ times more than CO2 fluxes (please see our reply to the 1st comment of 2nd reviewer). If we increase the CH4 prior uncertainty, this number will become even higher making the CO2 fluxes too restricted.

We have made the following update to our manuscript (Discussion Section) to address the optimistic estimates of posterior uncertainties:

It is noteworthy that the inversions are run assuming uncorrelated measurements and a perfect transport. Also, as we are not optimizing the atmospheric sink of CH4, all the information from its budget is used to constrain the surface fluxes. Hence, the estimates of posterior uncertainties tend to be optimistic in this study. The $\chi^2$ statistic indicates whether the assumed measurement and prior errors are statistically consistent (Meirink et al., 2008). We find $\chi^2/n_s = 0.93$ for RATIO, 0.96 for PR-CT, 0.93 for PR-LM and 1.14 for SURF in the CH4 inversions ($n_s$ is the number of observations assimilated in the inversion). This shows that we are not drastically underestimating the prior uncertainties in our CH4 inversions.

**Minor Comment**:

- *P2 L5: still consistent results are obtained. Consistent with respect to what?*

still consistent results are obtained with respect to other CH4 inversions.

- *P2 L10: original information? Im not sure what this means.*

Atmospheric measurements of GHGs can provide information about their atmospheric budget. Inverse modeling methods, also known as top-down approaches, have been developed to make use of this information to obtain improved estimates of surface fluxes

- *P4, Section 2: This could do with a brief overview of the chemical transport model setup (fluxes, OH fields, dynamics, etc).*

AC: we have added the following to our manuscript (in method section):

We use the TM5-4DVAR inversion modeling system. It is comprised of the Tracer Transport Model version 5 (TM5, Krol et al., 2005) coupled to a variational data assimilation system (4DVAR, Meirink et al., 2008). TM5 simulates the spatiotemporal distribution of a tracer in the atmosphere for a given set of fluxes. In this study, TM5 is run at a 64 degree horizontal resolution and 25 vertical hybrid sigma pressure levels from the surface to the top of the atmosphere. The meteorological fields for this offline model are taken from the European Centre for Medium-Range Weather Forecasts (ECMWF) ERA-interim reanalysis [Dee et al., 2011].

- *P5 L4:7. Are these choices largely subjective? Reasons should be given, and a discussion of the implications. See general comment above.*

AC: Please refer to our reply to the general comment above.

- *P5 L14: Missing full stop before The RemoTeC*

- *P5 L18: referred to as*

- *P5 L25: Use of additional here. . . additional to what (I presume it means in addition to the TCCON bias correction, which is described in the section below).*

- *P8 L1: Use of uncorrelated observation/model errors. Is it realistic to assume that the observation and model representation errors are uncorrelated in space and time? What might be the implications of this choice?*

AC: A Bayesian inverse model should in principle address the correlation of observations and weight them properly. The observations are assimilated in the inversion by comparing them with model-simulated mixing ratios. Therefore, errors in the model can also create a correlation. In practice, correlations are often ignored, both because they are difficult to quantify, and properly taking them into account slows down the inversion systems by a large extent. A prominent illustration of this is given by the numerical weather prediction systems since most of them assume uncorrelated

observation errors (but correlated prior errors). Inverses modeling studies of CH4 dont take them into account directly (for example, Alexe et al., 2014, Houweling et al., 2014, Monteil et al., 2013). Studies assimilating SCIAMACHY measurements implemented the binning method to reduce the impact of clustered measurements of SCIAMACHY (Houweling et al., 2014, Monteil et al., 2013). However, this is not so critical for GOSAT, as the number of soundings is lesser in amount. Also, the main goal of this study is to compare the results of different inversion methods in a consistent setup. All the inversions are done assuming uncorrelated observation/model errors, so this should not affect our results drastically.

*- P8 L18: Re-word relatively less errors is not grammatically correct.*
*- P11 L9: do not show an important seasonal dependence. This needs re-wording. I dont understand what an important seasonal dependence means.*
*- P11 L18: information that is used (remove comma).*
*- P17 L18: poorer rather than lesser*
*- P21 L4: Isnt it fairer to say that they largely cancel out? The cancellation is not 100%.*
*- P23 L8: Isnt 5x5 degrees quite a large area? Can this analysis be carried out over smaller scales?*

AC: 5x5 degrees area provides us with a sufficient number of GOSAT retrieval around the TCCON sites.

AC: All other minor comments are addressed in the revised manuscript.

AC: We agree with the reviewer, however, there are some limitations to the calculation done above. In our inversions, we do not optimize anthropogenic CO2 fluxes. Also, the fluxes are weighted with their respective error in the cost function. We calculate the same number by adding a $1\sigma$ perturbation of global CH4 ($\approx$15Tg/yr) and CO2 (1.2GtC/yr) fluxes in the atmosphere. This will have the corresponding change in the mixing ratio of the tracers in the atmosphere. The change in CH4 mixing ratio will be (1800 ppb x (15 Tg.yr-1/5000 Tg))= 5.4 ppb.yr$^{-1}$ = 0.3%.yr$^{-1}$ . For CO2, it will be (400ppm x 1.30 PgC.yr$^{-1}$/860 PgC)= 0.6 ppm.yr$^{-1}$ = 0.15 %.yr$^{-1}$. So the Xratio will be impacted 0.3/0.15  2 times more due to CH4 than CO2.

As the inversion is adjusted with the square of the observations, CH4 fluxes will be adjusted $2^2 = 4$ times more than CO2 fluxes. However, this number can be different on regional scales. As CO2 and CH4 surface data are assimilated also and both of them receive approximately equal weight in the inversion, the ratio may be lower. We have added the following to the revised manuscript:

The signal from Xratio can be up to $\approx 4$ times more sensitive to adjustment of CH4 fluxes than CO2 fluxes in our inversion setup. In such case, the surface observations of CO2 become the primary constraint on the CO2 fluxes. This can be further verified by looking at Supplementary figure 2, in which the RATIO and SURF inversion show very good agreement. It should be also noted that latitude dependent XCH4 bias contributed by the combination of transport model and retrieval biases become more important in a Xratio inversion while errors reduce due to aerosol and cloud scattering.

**Technical corrections:**

*Page 01- Line 03 Putting here "biased" instead of "heavily biased" would suffice, referring to current state of retrievals.*

*02-17 and 02-25 Lists of papers are similar, likely to present same information twice, better to put some distinction. Adding Deng et al ACP 2014 and Maksyutov et al ACP 2013 may be useful for completeness.*

*02-31 two types of retrieval methods can be used in place of two retrieval methods*

*03-21 (Fraser et al., 2014) -¿ Fraser et al., (2014)*

*04-27 As- sessment -¿ Assessment*

*04-26 right spell should be v.4.2 FT2010*

*05-01 onJacobson -¿ on Jacobson*

*05-12 adding reference to Remotec (Butz?) would help here.*

*08-04 GOSAST -¿ GOSAT.*

*08-20 Should ppm/ppm be used in place of ppb/ppm?*

*09-01 Units of table 1 need more explanation. Text says it is percentage difference weighted with GOSAT+TCCON error, the value doesnt look like percentage.*

*25-01 inChevallier -¿ in Chevallier*

*28-31 In the reference list initials like A. are appearing as a. in multiple locations.* AC: All technical corrections are addressed in the revised manuscript.

**1.2.1 References**

Randerson, J. T. and coauthors, Substrate limitations for heterotrophs: Implications for models that estimate the seasonal cycle of atmospheric CO2, Global Biogeochem. Cycles, 10(4), 585602, doi:10.1029/96GB01981, 1996.

Melton, J. R., and coauthors, Present state of global wetland extent and wetland methane modelling: conclusions from a model inter-comparison project (WETCHIMP), Biogeosciences, 10, 753-788, doi:10.5194/bg-10-753-2013, 2013.

[revised manuscript text omitted]

We use the TM5-4DVAR inversion modeling system. It is comprised of the Tracer Transport Model version 5 (TM5, Krol et al. (2005) ) coupled to a variational data assimilation system (4DVAR, Meirink et al. (2008) ). TM5 simulates the spatiotemporal distribution of a tracer in the atmosphere for a given set of fluxes. In this study, TM5 is run at a $6^o \times 4^o$ degree horizontal resolution and 25 vertical hybrid sigma pressure levels from the surface to the top of the atmosphere. The meteorological fields for this

offline model are taken from the European Centre for Medium-Range Weather Forecasts (ECMWF) ERA-interim reanalysis Dee et al. (2011) .

[revised manuscript text omitted]

It is noteworthy that the inversions are run assuming uncorrelated measurements and a perfect transport. Also, as we are not optimizing the atmospheric sink of $CH_4$, all the information from its budget is used to constrain the surface fluxes. Hence, the estimates of posterior uncertainties tend to be optimistic in this study. The $\chi^2$ statistic indicates whether the assumed measurement and prior errors are statistically consistent (Meirink et al., 2008). We find $\chi^2/n_s = 0.93$ for RATIO, 0.96 for PR-CT, 0.93 for PR-LM and 1.14 for SURF in the $CH_4$ inversions ($n_s$ is the number of observations assimilated in the inversion). This shows that we are not drastically underestimating the prior uncertainties in our $CH_4$ inversions.

[revised manuscript text omitted]